# Erythroid differentiation regulator-1 induced by microbiota in early life drives intestinal stem cell proliferation and regeneration

Hirohito Abo[1], Benoit Chassaing [1,2,3,4], Akihito Harusato [1], Miguel Quiros[5], Jennifer C. Brazil[5], Vu L. Ngo[1], Emilie Viennois[6], Didier Merlin[6,7], Andrew T. Gewirtz[1], Asma Nusrat[5] & Timothy L. Denning[1]*

Gut microbiota and their metabolites are instrumental in regulating intestinal homeostasis. However, early-life microbiota associated influences on intestinal development remain incompletely understood. Here we demonstrate that co-housing of germ-free (GF) mice with specific-pathogen free (SPF) mice at weaning (exGF) results in altered intestinal gene expression. Our results reveal that one highly differentially expressed gene, erythroid differentiation regulator-1 (Erdr1), is induced during development in SPF but not GF or exGF mice and localizes to Lgr5[+] stem cells and transit amplifying (TA) cells. Erdr1 functions to induce Wnt signaling in epithelial cells, increase Lgr5[+] stem cell expansion, and promote intestinal organoid growth. Additionally, Erdr1 accelerates scratch-wound closure in vitro, increases Lgr5[+] intestinal stem cell regeneration following radiation-induced injury in vivo, and enhances recovery from dextran sodium sulfate (DSS)-induced colonic damage. Collectively, our findings indicate that early-life microbiota controls Erdr1-mediated intestinal epithelial proliferation and regeneration in response to mucosal damage.

[1] Center for Inflammation, Immunity & Infection, Institute for Biomedical Sciences, Georgia State University, 100 Piedmont Ave, Atlanta, GA 30303, USA. [2] Neuroscience Institute and Institute for Biomedical Sciences, Georgia State University, Atlanta, Georgia, USA. [3] INSERM, U1016, Paris, France. [4] Université de Paris, Paris, France. [5] Department of Pathology, University of Michigan, Ann Arbor, MI 48109, USA. [6] Center for Diagnostics and Therapeutics, Institute for Biomedical Sciences, Georgia State University, Atlanta, GA 30303, USA. [7] Atlanta Veterans Affairs Medical Center, Decatur, GA 30033, USA. *email: tdenning@gsu.edu

The intestinal microbiota is comprised of hundreds of trillions of microbes encompassing bacteria, viruses, fungi, protists, and archae that co-develop with the host from very early in life[1]. Components of the intestinal microbiota influence mucus production, epithelial and sub-epithelial cellular dynamics, immune system development, and promote and maintain homeostasis in the intestine[2–8]. Alterations in the composition of the intestinal microbiota are linked to the development of numerous intestinal and extra-intestinal disease manifestations, including inflammatory bowel disease, asthma, obesity, metabolic syndrome, and diabetes[9–12]. These effects can result from direct microbiota–cellular interactions and indirect effects associated with microbiota-derived metabolites[13].

The maternal microbiota begins to colonize the host at birth and molecules derived from the microbiota can be transferred to the neonate through maternal milk[14,15]. Maternal microbiota can also influence the developing fetus via placental exchange of microbial molecules in utero. Metabolism of dietary components and xenobiotics by the maternal microbiota can additionally enter the mother's serum and subsequently reach the fetus. While the effects of nutrition, alcohol, and medications on fetal development are well-appreciated, the understanding of the maternal microbiota influences on fetal development and associated health and disease outcomes into adulthood remains limited.

The relationship between the microbiota and immune system development is driven in part by postnatal effects of the neonatal microbiota[3,14,16]. For example, the dramatic increase in microbiota around the time of weaning in mice coincides with the expansion of Foxp3+ regulatory T cells and rise in IgA[17]. These important immunologic changes involve microbiota-dependent generation of short-chain fatty acids and the vitamin A metabolite, retinoic acid[18]. Intestinal immune system development is further under the control of the maternal microbiota during pregnancy. Using an elegant experimental model of gestation-only colonization of germ-free (GF) mice, transient maternal colonization with *E. coli* is sufficient to drive postnatal increases in intestinal group 3 innate lymphoid cells and F4/80+CD11c+ mononuclear cells, but not adaptive immune cells, in the offspring[14]. Significant transcriptional changes are also observed among signature genes for specific epithelial lineages and functions. Collectively, these data highlight an emerging role for early-life microbiota in controlling immune and intestinal epithelial barrier defense.

The intestinal epithelial barrier is instrumental in the physical separation of the microbiota from the host. This single layer epithelium is self-renewed and continuously replaced every 2–5 days and this process is tightly orchestrated by intestinal stem cells (ISCs)[19]. Leucine-rich repeat-containing G protein-coupled receptor 5 (Lgr5)-expressing ISCs give rise to highly proliferative transit amplifying (TA) cells, which differentiate into all epithelial lineages including Paneth cells, tuft cells, enteroendocrine cells, goblet cells, and enterocytes along crypt-villus axis[20]. This differentiation process can be influenced by the microbiota and microbial metabolites, as evidenced by elongation of villi and shortening of crypts in GF rodents. Notably, some changes present in the epithelium are reversible by re-conventionalization, whereas other changes are long-lasting suggestive of epigenetic regulation[14,15].

In this study, we explore the contribution of early-life microbiota to enduring changes in intestinal gene expression. We employ an experimental model wherein mice are born GF and subsequently colonized with specific-pathogen-free (SPF) microbiota at the time of weaning (exGF). Using this model, we focus on transcriptional changes in adult exGF mice that are irreversible by colonization with SPF microbiota. We identified one of the top-most differentially expressed genes between SPF and exGF mouse small intestine and colon to be erythroid differentiation regulator-1 (Erdr1), a soluble factor that regulates cellular survival, metastasis, and NK-mediated cytotoxicity[21,22]. Our findings show that Erdr1 is induced by microbiota in early life, localizes to Lgr5+ ISCs and TA cells and induces intestinal epithelial proliferation and regeneration in response to mucosal damage.

## Results

**Early-life microbiota regulates Erdr1 expression.** To determine the effects of early-life microbiota exposure on intestinal gene expression, we employed an experimental model system using SPF, exGF, and GF mice. exGF mice were born and raised in GF conditions until weaning (day 21) at which time they were transferred into SPF conditions and cohoused with age/sex-matched mice until day 42 (Fig. 1a). In order to evaluate intestinal gene expression difference between SPF, exGF and GF mice, we performed RNA sequencing using total small and large intestinal tissue at day 42. Volcano plot analysis of RNAseq data revealed *Erdr1* as one of the top genes categorized as down in exGF or down in GF indicating preferential expression in the small and large intestines of SPF mice when compared with exGF or GF mice (Fig. 1b; Supplementary Data 1). We next analyzed Erdr1 mRNA expression by quantitative real-time PCR (qPCR). In the duodenum, jejunum, ileum, and colon of SPF mice Erdr1 mRNA was detected, whereas expression in exGF and GF samples was undetectable (Fig. 1c). Given a previous report that the induction of Erdr1 was dependent upon Myd88 signaling in splenic CD4+ T cells[23], we therefore tested whether knockout of Myd88 influenced Erdr1 expression in the small and large intestine by qPCR. Results from these experiments indicate that Erdr1 expression is not affected by ablation of Myd88 (Supplementary Fig. 1).

To begin to gain insight into epigenetic modifications that were associated with altered Erdr1 expression between SPF, exGF, and GF groups, we investigated histone H3 acetylation (H3Ac), which is linked to gene activation[24]. Following chromatin immunoprecipitation (ChIP) with H3Ac antibody, we observed significant decreases in amplification of the Erdr1 promoter region in the large intestine of exGF and GF mice, relative to SPF mice (Fig. 1d), which is indicative of gene repression and consistent with the lack of Erdr1 expression in Fig. 1c. Collectively, these data demonstrate that early-life microbiota regulates intestinal Erdr1 expression in association with enhanced histone acetylation. Further, colonization of exGF mice with SPF microbiota at weaning was insufficient to induce Erdr1 suggesting a critical early-life window for the induction of Erdr1 by the microbiota.

**Erdr1 is expressed in Lgr5+ ISCs and TA cells.** Erdr1 was initially identified by screening the WEHI-3 cell line for secreted proteins that induce hemoglobin synthesis[25]. It is highly conserved between mice and humans and has been reported to be expressed in keratinocytes, pulmonary mesenchymal cells, murine embryonic fibroblasts, melanoma cells, and splenic T cells[23,26,27]. However, the expression and function of Erdr1 within the intestine has not been explored. To investigate Erdr1 expression in the intestine, we performed in situ hybridization using RNAscope[28]. As shown in Fig. 2a, Erdr1 RNA expression localized primarily in the crypt region and TA zone of the small and large intestine, with less prominent staining in the lamina propria and submucosal region, which is consistent with a previous report demonstrating Erdr1 expression in CD4+ T cells[23] (Supplementary Fig. 2). Consistent with qPCR analysis in Fig. 1c, Erdr1 RNA was undetectable in the small and large intestines from exGF and GF mice.

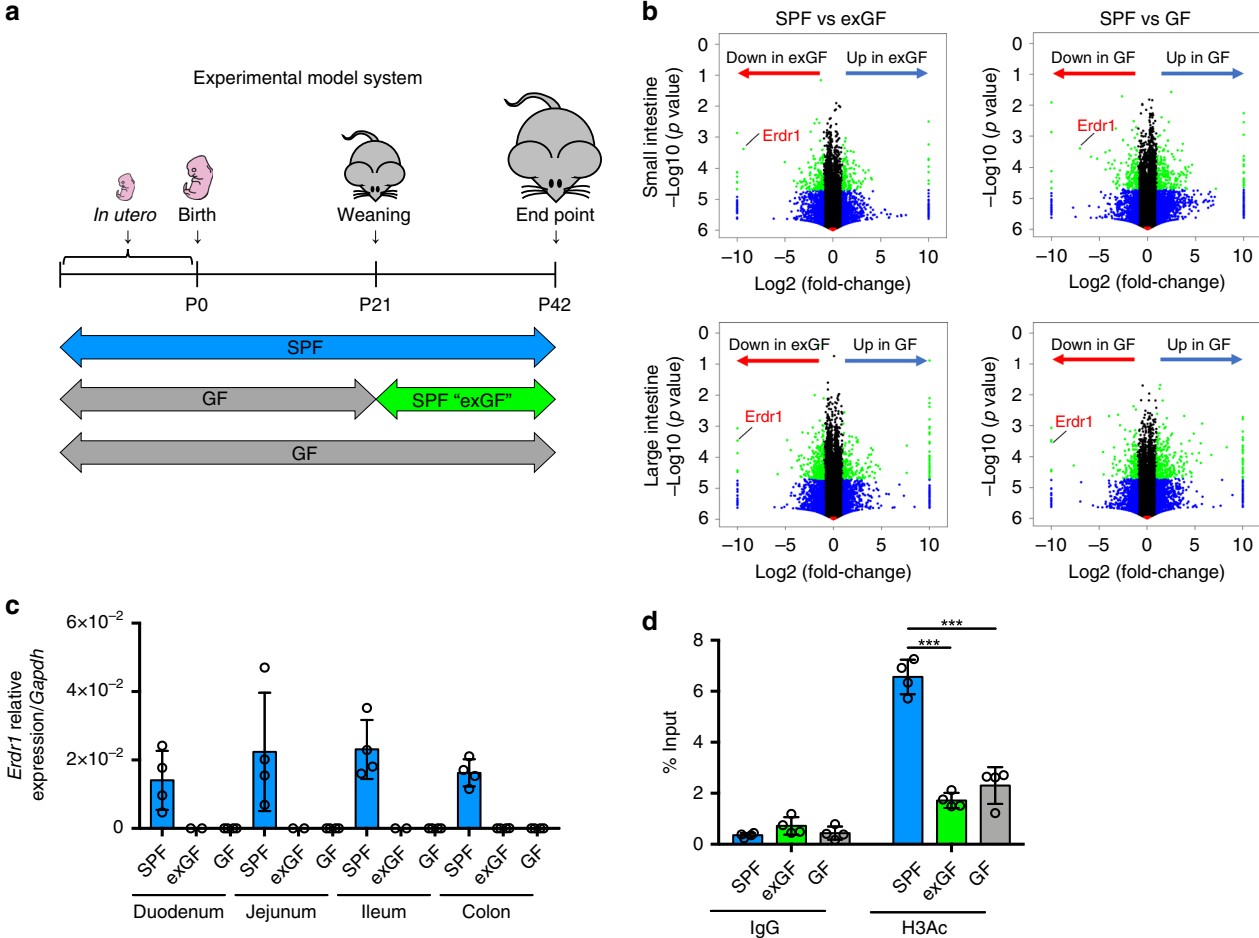

**Fig. 1 Early-life microbiota regulates Erdr1 expression. a** Experimental schematic of the exGF model: Germ-free (GF) mice were colonized with microbiota beginning at day 21 by co-housing with SPF mice for 3 weeks. **b** Volcano plots showing log2 fold-change of total colon gene expression in SPF mice compared with exGF or GF mice. Genes that were significantly decreased in exGF or GF compared with SPF colons are highlighted on red, while those increased in exGF or GF are highlighted in blue. **c** qPCR analysis of *Erdr1* using whole small and large intestinal tissue from SPF, exGF, and GF mice. $n = 2$ (duodenum/jejunum/ileum from exGF), $n = 4$ (duodenum/jejunum/ileum from SPF and GF, colon). **d** ChIP analysis for histone H3 acetylation (H3Ac) in the Erdr1 promoter region using total colon tissue. $n = 4$; two independent experiments. All data are presented as mean ± SD; $*P < 0.05$, $**P < 0.01$, $***P < 0.001$ by one-way ANOVA with Tukey's multiple comparison test.

The intestinal crypt region is highly dynamic and comprised of several cell types, including Lgr5[+] ISCs, Paneth cells, and goblet cells[29]. To further define which cells in the crypt region expressed Erdr1, we performed multicolor RNAscope on sections of the small intestine using the ISC marker Lgr5, and the Paneth cell marker Lyz1[30]. Using this approach, Erdr1 RNA expression was observed to co-localize with Lgr5, but not Lyz1, as shown in longitudinal sections (Fig. 2b) and transverse sections (Fig. 2c). Collectively, these findings demonstrate that Erdr1 is expressed by Lgr5[+] ISCs and TA cells in SPF mice, but not exGF or GF mice.

**Erdr1 increases growth of intestinal organoids.** Previous studies have reported that Erdr1 is involved in NK cell activation[31], T-cell apoptosis[23], and melanoma migration[26]. However, the role of Erdr1 in the intestine remains unknown. Based on the expression of Erdr1 by intestinal epithelial cells, especially Lgr5[+] ISCs and TA cells, we explored the effects of recombinant Erdr1 (Supplementary Fig. 3) on in vitro intestinal organoid cultures using standard organoid medium containing EGF, Noggin, and R-spondin-1[32]. We first examined the expression of endogenous Erdr1 in organoid cultures from SPF, exGF, and GF mouse small

and large intestines. Similar to results from total intestinal tissue directly ex vivo, we observed Erdr1 mRNA expression in both small and large intestinal organoids derived from SPF, but not exGF or GF mice (Fig. 3a). In light of these data, we next explored the H3Ac status of Erdr1 in organoid cultures using ChIP. Consistent with observations directly ex vivo using total intestinal tissue (Fig. 1d), H3Ac of the Erdr1 promoter region was higher in organoids derived from SPF mice when compared with those derived from exGF and GF mice (Fig. 3b). Further, organoid efficiency, budding, and surface area in small and large intestinal organoids were significantly greater in SPF organoids when compared with exGF or GF organoids. We also observed that the addition of Erdr1 resulted in enhanced growth of SPF organoids, and reversed the impaired growth of exGF and GF organoids (Fig. 3c–f, Supplementary Fig. 4a–d). To ensure that these effects were not a result of LPS contamination in recombinant Erdr1, we performed similar studies using organoids generated from TLR4-deficient mice and observed nearly identical results as in wild-type controls (Supplementary Fig. 5a–c). In agreement with the ability of Erdr1 to increase organoid efficiency, budding and surface area, cell cycle analysis revealed that the addition of Erdr1 to large intestine organoid cultures increased the percentage of

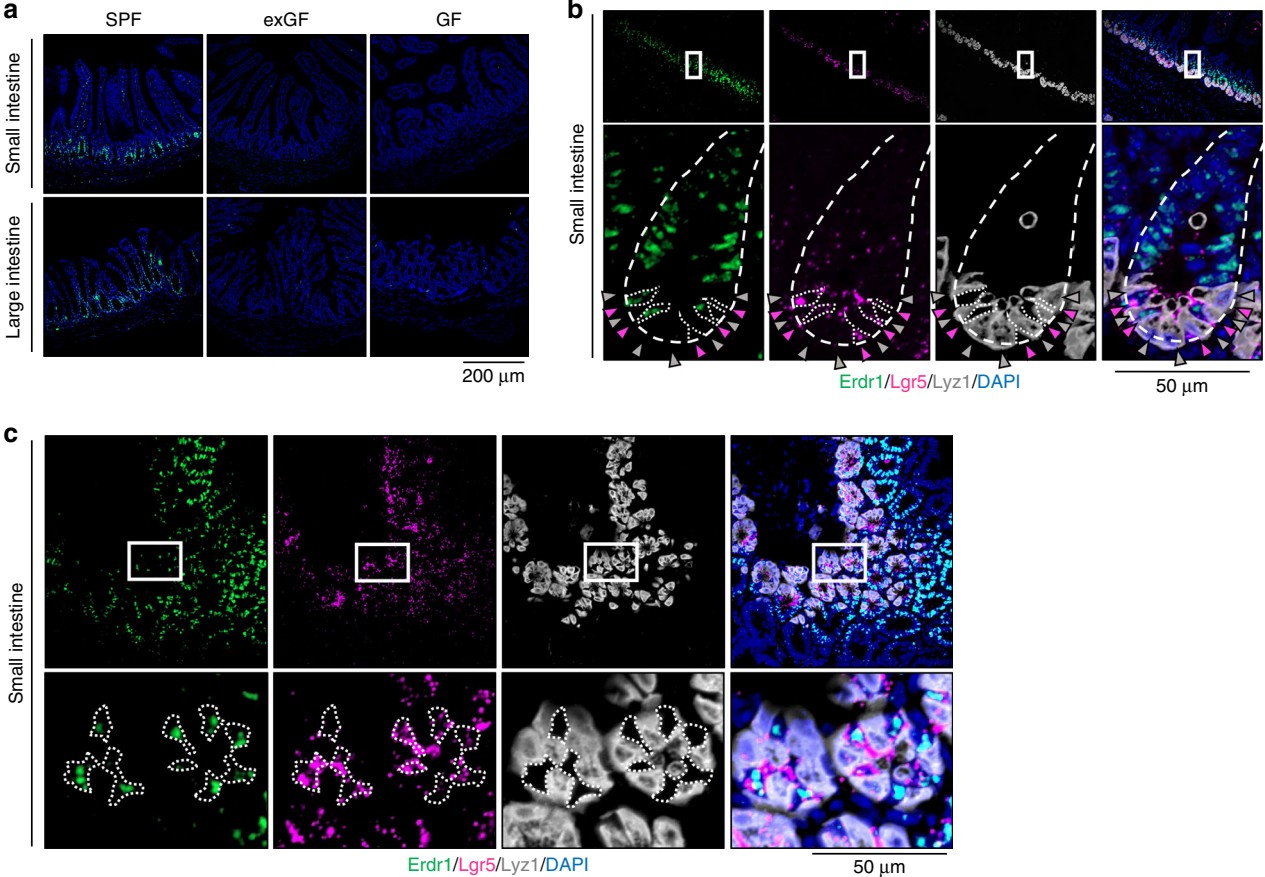

**Fig. 2 Erdr1 is expressed in Lgr5$^+$ intestinal stem cells (ISCs) and transit amplifying (TA) cells. a** RNAscope staining for *Erdr1* within small and large intestine from SPF, exGF, and GF mice. Green indicates *Erdr1* RNA. **b** 3-color RNAscope staining of longitudinal sections of small intestine: green, Erdr1; pink, Lgr5; white, Lyz1; blue, DAPI. **c** 3-color RNAscope staining of transverse sections of small intestine, as in **b**.

cells in S and G2/M phase, while the percentage of cells in the G0/G1 phase was decreased (Supplementary Fig. 6). Since previous study revealed that Erdr1 promotes T-cell apoptosis[23], we investigated whether Erdr1 induce apoptosis in intestinal organoid. Using Annexin V which is well-established cell marker for pre-apoptosis and PI, we observed no change of pre-apoptotic cells (Annexin V$^+$ PI$^-$) (Supplementary Fig. 7). Collectively, these data demonstrate that Erdr1 functions to enhance intestinal organoid growth.

**Erdr1 induces expansion of Lgr5$^+$ ISCs.** Lgr5$^+$ ISCs are critical components of the crypt niche that divide to generate stem cells and TA daughter cells that further give rise to terminally differentiated progenies including absorptive enterocytes, secretory goblet, entero-endocrine, tuft and Paneth cells[33]. To evaluate the effect of Erdr1 on Lgr5$^+$ ISCs, we performed qPCR analysis of ISC signature genes on small and large intestinal organoids cultured in the presence or absence of Erdr1. Following Erdr1 stimulation of organoids generated from SPF mice for 6 days, the expression of ISC signature genes (*Lgr5*, *Olfm4*, *Smoc2*, *Ascl2*, and *Tnfrsf19*) was significantly increased in large intestine (Fig. 4a) and small intestine organoids (Supplementary Fig. 8) when compared with untreated control organoids. Consistent with these findings, we detected increased frequency and number of Lgr5-EGFP$^+$ cells in small intestine organoids derived from Lgr5-reporter mice in which Lgr5 expressing cells are directly tagged with EGFP (Fig. 4b–d). Taken together, these data demonstrate that Erdr1 increases ISC signature genes and enhances Lgr5$^+$ ISC expansion.

**Erdr1 enhances Wnt signaling in IECs and organoids.** Wnt/β-catenin signaling plays an essential role in Lgr5$^+$ ISC maintenance and differentiation[34]. Conditional deletion of Tcf4, the key effector of Wnt signaling pathway, was shown to result in rapid loss of Lgr5$^+$ ISCs[35] and a similar phenotype was observed using organoids where Wnt3 was specifically deleted in intestinal epithelial cells[36]. To determine whether Erdr1 induces Wnt/β-catenin signaling, we performed a TOP/FOP luciferase reporter assay using the intestinal epithelial cell line SKCO15. Using this approach, we observed dramatically increased luciferase reporter activity (approximately sixfold) with two different concentrations of Erdr1 (Fig. 5a). Furthermore, we observed enhanced expression of the well-established Wnt pathway targets genes including *Myc*, *Axin2*, *Sox9*, and *Ephb2* within large intestine (Fig. 5b) and small intestine organoids (Supplementary Fig. 9). Consistent with these findings, we also detected translocation of activated β-catenin to the nuclei within large intestinal organoid cultures (Supplementary Fig. 10). Furthermore, we found no change in the expression of the Notch signaling genes *Hes1*, *Yap1* and their target genes *Cyr62* and *Ctgf*, which are associated with intestinal organoid growth (Supplementary Fig. 11a, b)[37,38]. Collectively, these results demonstrate that Erdr1 enhances Wnt signaling in intestinal epithelial cells and organoids.

**Erdr1 accelerates wound closure in mouse and human IECs.** Given that Erdr1 is expressed by Lgr5$^+$ ISCs and TA cells that are in close juxtaposition with enterocytes, we next explored the effects of Erdr1 on intestinal epithelial cells using an in vitro scratch-wound assay. Defined scratch wounds were created in

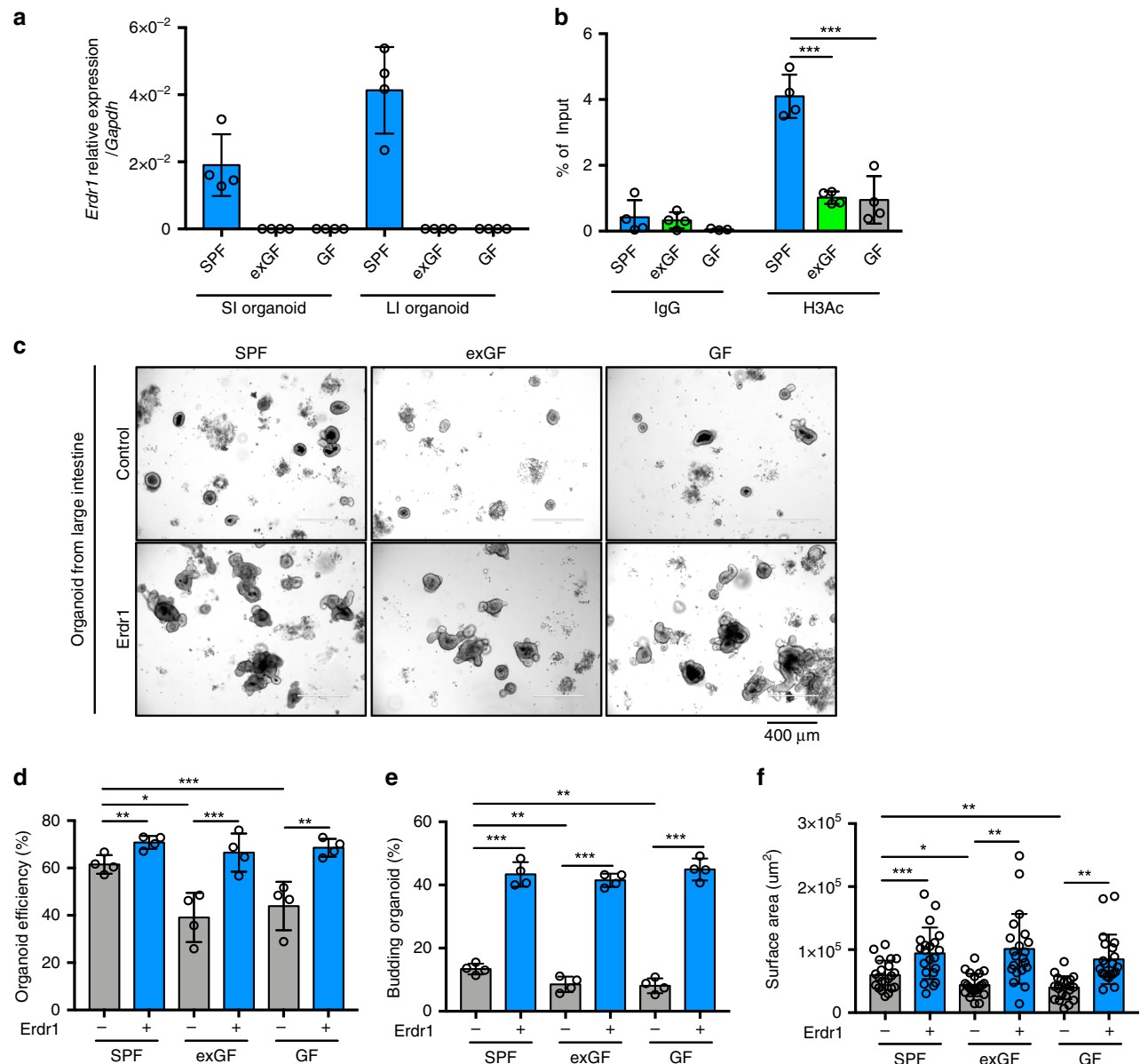

**Fig. 3 Erdr1 increases growth of intestinal organoids. a** Expression of *Erdr1* in intestinal small and large organoids derived from SPF, exGF and GF mice. $n = 4$. **b** ChIP analysis for histone H3 acetylation (H3Ac) in the Erdr1 promoter region using large intestine (LI) organoids derived from SPF, exGF, and GF mice. $n = 4$; two independent experiments. **c** Representative images of LI organoids derived from SPF, exGF, and GF mice and cultured ±Erdr1 (1 µg) for 6 days. **d** Organoid efficiency of LI organoids cultured ±Erdr1. $n = 4$; representative of three experiments. **e** Frequency of budding organoids cultured ±Erdr1 for 6 days. $n = 4$; representative of three experiments. **f** Surface area of LI organoids cultured ±Erdr1. $n = 21$; representative of three experiments. All data are presented as mean ± SD; *$P < 0.05$, **$P < 0.01$, ***$P < 0.001$ by unpaired, two-tailed *t*-tests.

monolayer cultures using the small intestinal epithelial cell line Mode-K, and then wound closure was assessed in the presence and absence of Erdr1 supplementation. As shown in Fig. 6a, b and Supplementary movie 1, the addition of Erdr1 accelerated wound closure in Mode-K cells. Cell cycle analysis further revealed that Erdr1 increased G2/M and S phase cells and decreased the G0/G1 phase cells (Fig. 6c, d). Consistent with these data, staining with the proliferation marker Ki-67 was increased by Erdr1 (Fig. 6e–g). Alternatively, knockdown of endogenous Erdr1 using siRNA (Supplementary Fig. 12) reduced wound closure in Mode-K cells (Fig. 6h, i). Since Erdr1 is highly conserved between mice and humans[25], we next investigated the effects of Erdr1 on human intestinal epithelial cells. We performed wound closure assays and cell cycle analyses using the human colon cancer cell line HT-29 and observed similar

findings as with Mode-K cells that Erdr1 accelerated wound closure and increased G2/M and S phase cells and decreased the G0/G1 phase cells (Supplementary Fig. 13a–d). In addition, used human intestinal organoids that were differentiated into a 2D monolayer and then subjected to defined scratch wounds and cultured with or without Erdr1. As with Mode-K and HT-29 cells, we observed that human 2D organoid monolayers treated with Erdr1 showed enhanced wound closure when compared with controls (Supplementary Fig. 13e, f). Taken together, these results demonstrate that Erdr1 stimulates in vitro intestinal epithelial wound closure in mouse and human cells.

**Erdr1 promotes the regeneration of Lgr5+ ISCs.** It is appreciated that proliferating cells include Lgr5+ ISCs and TA cells are

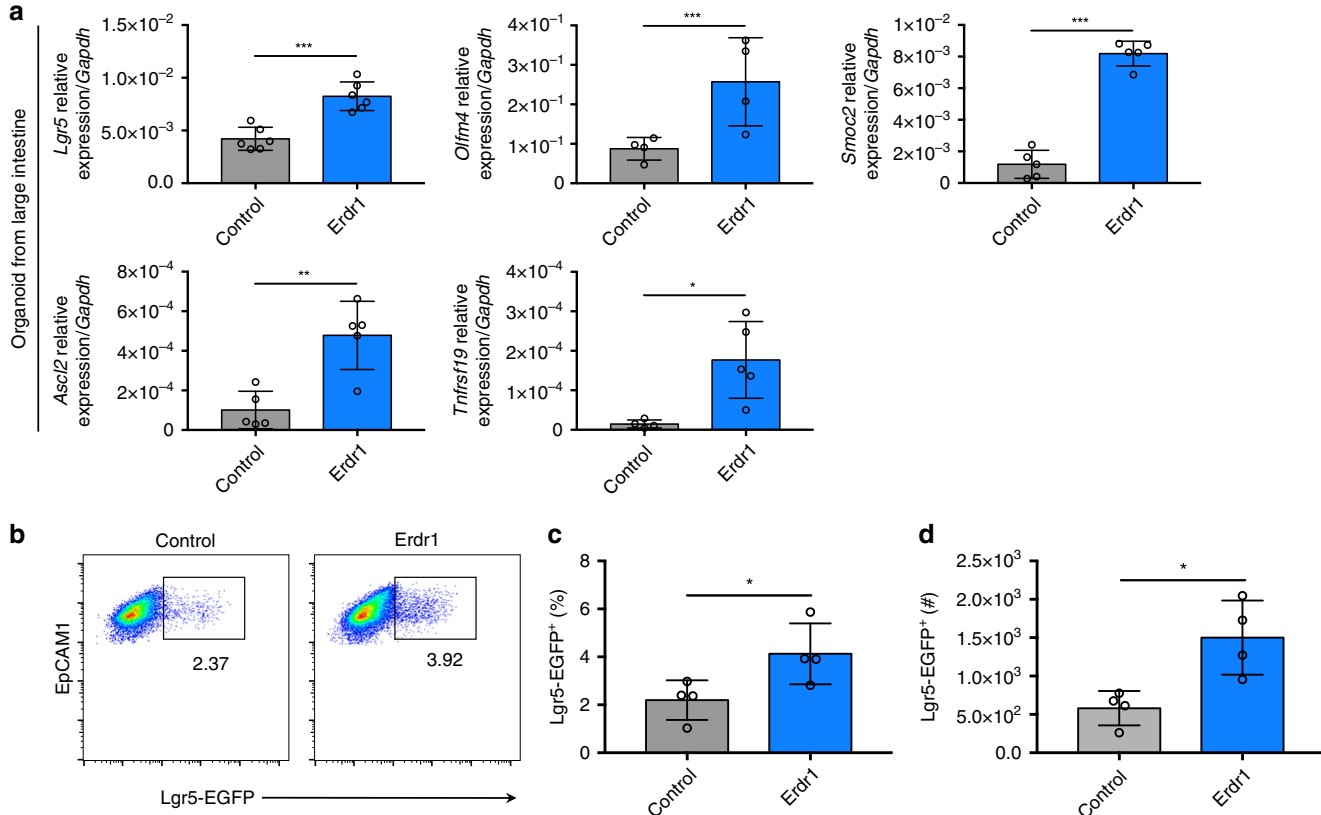

**Fig. 4 Erdr1 induces ISC signature gene expression and Lgr5$^+$ ISCs. a** Expression of stem cell signature genes (*Lgr5*, *Olmf4*, *Smoc2*, *Ascl2*, *Tnfrsf19*) in LI organoids from SPF mice cultured ±Erdr1 for 6 days. $n = 4$ (*Olfm4*), $n = 5$ (*Smoc2*, *Ascl2*, *Tnfrsf19*), $n = 6$ (*Lgr5*); representative of three experiments. **b–d** Frequency and cell numbers of Lgr5-EGFP$^+$ cells (live cells, EpCAM1$^+$, EGFP$^+$) in SI organoids cultured ±Erdr1 for 6 days. $n = 4$; representative of three experiments. All data are presented as mean ± SD; *$P < 0.05$, **$P < 0.01$, ***$P < 0.001$ by unpaired, two-tailed *t*-tests.

highly sensitive to DNA damage caused by radiation[39], and the self-renewal capacity of Lgr5$^+$ ISCs is key to recovery[40,41]. Based on our findings that Erdr1 increased the number of Lgr5$^+$ ISCs and promoted TA cell proliferation in the steady state, we explored whether Erdr1 could mediate similar effects in vivo following radiation-induced injury. Lgr5-EGFP reporter mice were irradiated (IR) and injected with Erdr1 daily to SPF mice. As expected, we observed a dramatic reduction in the frequency of Lgr5-EGFP$^+$ cells in the ileum of IR mice on day 3 post irradiation, and this reduction was largely reversed by the administration of Erdr1 (Fig. 7a, b). Similarly, reduced Ki-67$^+$ cells and BrdU$^+$ cells in both the ileum and colon at day 3 post irradiation was reversed by Erdr1 administration in time dependent manner (Fig. 7c, d, Supplementary Fig. 14a, b, and Supplementary Fig. 15a, b). Since radiation induces the apoptosis of intestinal crypt epithelial cells, we examined TUNEL staining to examine the effect of Erdr1 on apoptosis after radiation. In the control group, TUNEL positive cells was increased at day 1 post radiaton and decreased time dependent manner as well as Erdr1 treated group, which indicates Erdr1 does not have effect to apoptosis after radiation-induced injury (Supplementary Fig. 16a, b). These data indicate that Erdr1 can induce Lgr5$^+$ ISCs regeneration and proliferation of crypt cells following radiation-induced injury.

**Erdr1 promotes recovery from DSS-induced colitis**. Our findings indicating that Erdr1 promotes epithelial cell regeneration and proliferation in response to radiation-induced injury suggested that Erdr1 may also protect the intestinal epithelial barrier from other types of injury/damage. To explore this further, we employed the DSS model of colitis that exhibits intestinal

epithelial damage during DSS treatment followed by epithelial repair when DSS is discontinued and replaced with regular drinking water. Using this model system, we investigated whether delivery of Erdr1 or treatment with anti-Erdr1 antibody could influence recovery from DSS-induced colitis in vivo. Interestingly, the administration of Erdr1 to DSS-treated mice on day 4, 6, and 8 enhanced recovery as evidenced by reduced disease activity index scores (DAI; Fig. 8a), increased colon length (Fig. 8b, c), and reduced histology score (Fig. 8d, e). Alternatively, DSS-treated mice administrated anti-Erdr1 antibody on day 4 and 7 showed opposite effects—higher DAI scores, reduced colon length, and enhanced histology score (Fig. 8a–e). Collectively, these data establish that Erdr1 can play a beneficial role in recovery from DSS-induced colonic damage.

## Discussion

In this study we provide evidence demonstrating that Erdr1 is an early-life microbiota-inducible factor that functions to positively regulate intestinal barrier dynamics. Predominant expression of Erdr1 in the proliferative zone of the intestine, particularly among Lgr5$^+$ ISCs and TA cells, correlated with its ability to enhance Lgr5$^+$ ISC and organoid growth. Importantly, Erdr1 accelerated wound closure in vitro and promoted Lrg5$^+$ ISC regeneration and recovery following mucosal injury. Overall, these data highlight a fundamental contribution of Erdr1 to the maintenance and regeneration of ISCs and the intestinal epithelial barrier.

Interestingly, a recent report investigating mechanisms by which the microbiota regulates T-cell responses observed that Erdr1 expression was suppressed by the microbiota[22,23]. CD4$^+$ T cells isolated from GF mice were shown to have elevated Erdr1

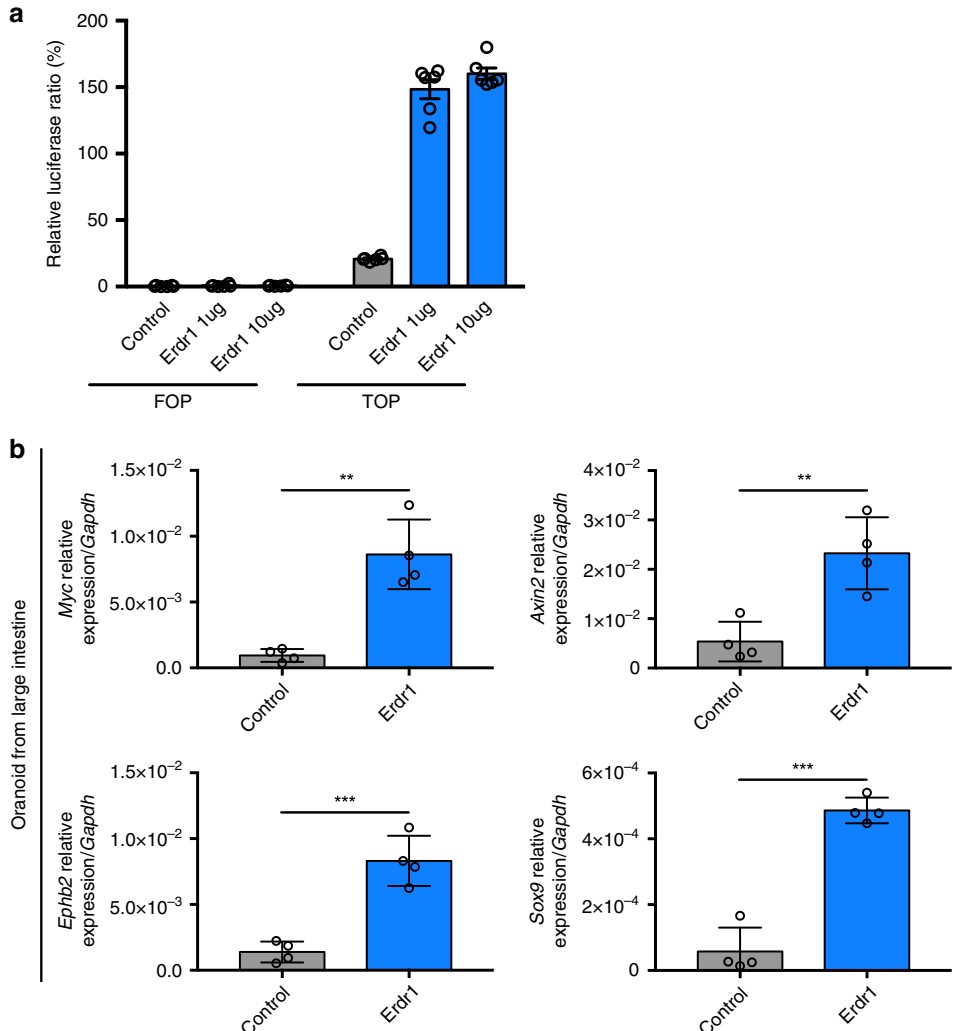

**Fig. 5 Erdr1 enhances Wnt signaling in intestinal epithelial cells and organoids. a** The relative luciferase activity measured by the TOP/FOP-flash assay in SKCO-15 human intestinal epithelial cells stimulated ±Erdr1. $n = 6$; representative of two experiments. **b** Expression of Wnt signaling pathway target genes (*Myc, Axin2, Ephb2, Sox9*) in LI organoid cultures ±Erdr1 (1 μg/mL). $n = 4$; representative of three experiments. All data are presented as mean ± SD; *$P < 0.05$, **$P < 0.01$, ***$P < 0.001$ by unpaired, two-tailed $t$-tests.

expression and enhanced apoptosis, while knockdown of Erdr1 resulted in enhanced T-cell survival. These study combined with our observations suggests highly dynamic temporal and cellular control of Erdr1 expression by the microbiota. While microbiota in early life may be promoting the expression of Erdr1 in Lgr5+ ISCs and TA cells, postnatal microbiota can regulate Erdr1 expression in CD4+ T cells. On the other hand, Erdr1 has been reported to be a survival signal during conditions of cellular stress[42]. Furthermore, Erdr1-expressing stroma can promote cancer cell survival in vitro and cancer cell invasion in vivo[26]. These findings indicate that the function of Erdr1 is highly cell dependent on cell type, condition and tissue. Clearly, additional studies are warranted to fully define how the microbiota controls immune and non-immune cells in the intestine and periphery.

Notably, Erdr1 expression in the intestine appears to be under strict temporal and perhaps epigenetic regulation. Even 3 weeks after exposure of GF mice to SPF conditions, intestinal Erdr1 was still not induced to levels approaching those observed in SPF mice. This suggests that if Erdr1 is not induced by the microbiota or microbial metabolites during a critical window in early life, gene modifications such as impaired histone H3Ac may restrict induction by the microbiota at later stages. Indeed, recent studies

have defined an important relationship between the microbiota and mammalian epigenetic pathways[24,43]. For example, microbiota-derived short-chain fatty acids have been shown to promote histone crotonylation in the colon via histone deacety-lases[44]. In addition, epithelial-specific deletion of HDAC3 results in increased susceptibility to *Citrobacter rodentium* in association with impaired IFN-γ production by CD8+ intraepithelial lym-phocytes[45]. Our findings of reduced histone H3Ac in the absence of microbiota in early life further underscores the relationship between the microbiota and epigenetic modification in the intestine. Importantly, our data do not rule out a role for other forms of epigenetic modifications, including methylation. The long-term consequences of these epigenetic changes regulating health and disease in the intestine in adulthood remain to be fully determined, but may play an important role in suppressing the development of colon cancer[46].

Our study definitively implicates early life microbiota in the induction of Erdr1 in the intestine; yet, it remains unclear if the critical window for Erdr1 induction is during in utero and/or early postnatal development. It is intriguing that Erdr1 is expressed in utero in the central nervous system beginning as early as embryonic day 11.5[47]. This suggest that Erdr1 may be an

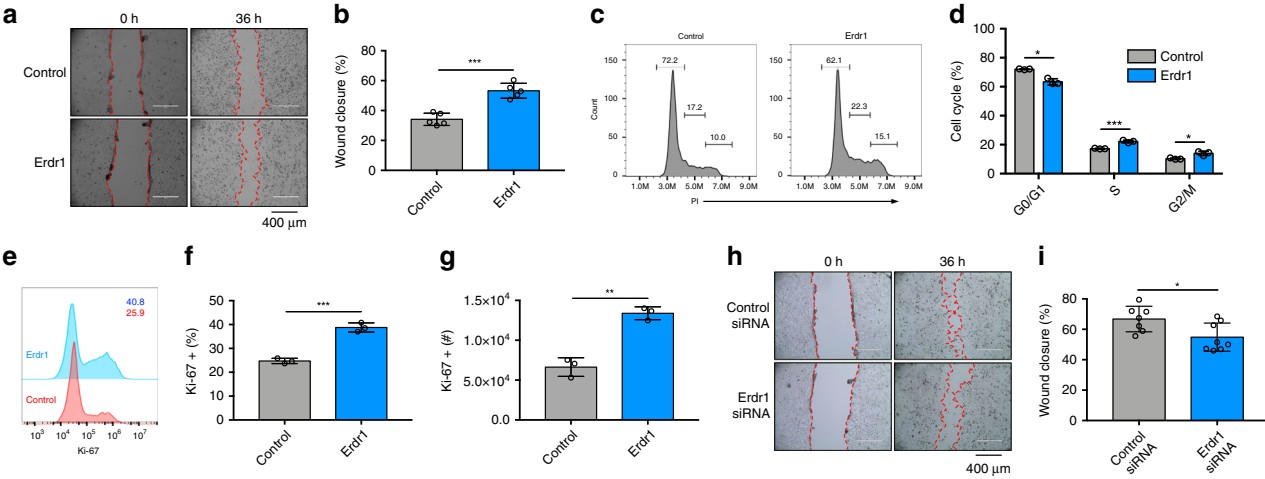

**Fig. 6 Erdr1 accelerates wound closure in mouse and human intestinal epithelial cells.** Scratch-wound assay using Mode-K mouse intestinal epithelial cells cultured ±Erdr1 (1 µg/mL) for 36 h (**a** picture; **b** % wound closure). $n = 5$; representative of three experiments. **c–d** Cell cycle analysis of Mode-K cells cultured ±Erdr1. $n = 3$; representative of three experiments (**c** histogram; **d** % cell cycle). **e–g** Ki-67 staining of Mode-K cells stimulated ±Erdr1. $n = 3$; representative of three experiments. **h–i** Wound closure of Mode-K cells treated with Erdr1 siRNA or control siRNA (**k** picture; **l** % wound closure). $n = 7$ (control siRNA), $n = 8$ (Erdr1 siRNA); representative of two experiments. All data are presented as mean ± SD; *$P < 0.05$, **$P < 0.01$, ***$P < 0.001$ by unpaired, two-tailed $t$-tests.

important factor induced during the development in numerous tissues under the control of diverse elements, including the microbiota. Consistent with this concept are data indicating that attempts to knockout Erdr1 in mice are not tolerated in utero[23]. Since the unborn fetus is believed to be in a sterile environment with initial microbial colonization happening at the time of birth, the induction of Erdr1 in the fetal intestine is not likely to be the result of direct interactions between the microbiota and fetus. During development in utero, however, the fetus is exposed to a plethora of metabolites from the mother's microbiota that pass through the placenta. This indirect influence of the microbiota on fetal development can have major influences on the intestine including increasing F4/80[+]CD11c[+] mononuclear cells and group 3 innate lymphoid cells, and increasing expression of genes encoding antimicrobial peptides and metabolism of microbial molecules[14,15]. Future studies employing transient colonization studies[14] and/or cross-fostering immediately after birth may help to shed light into how microbiota and microbial metabolites in the prenatal and early postnatal phases of development influences the expression of Erdr1 and other important genes in the intestine and periphery.

Beyond defining the importance of early-life microbiota in regulating Erdr1 expression, our results provide insight into the functions of this factor in epithelial proliferation and regeneration following injury. While relatively little remains known about the regulation of the ISC compartment after tissue damage, interleukin (IL)-22 has emerged as a critical factor in this process[48,49]. IL-22-mediated STAT3 phosphorylation in Lgr5[+] ISCs was shown to be central for intestinal organoid growth and epithelial regeneration[48] and more recently was reported to protect against genotoxic stress[49]. Although the receptor for Erdr1 has not yet been defined, the ability of Erdr1 to be produced by Lgr5[+] cells and also drive their expansion in association with activating the Wnt signaling pathway is somewhat analogous of the effects of R-spondin1 in intestinal organoid cultures[50]. The key function of R-spondin1 in organoids is to maintain a high activity of Wnt signaling and maintain Lgr5[+] ISC self-renewal capacity. Similarly, our studies demonstrate that Erdr1 enhanced Lgr5[+] ISCs in organoids. However, the effects of Erdr1 are not limited to Lgr5[+] ISCs and extend to fully differentiated intestinal epithelial cells in both the steady state and following injury. The effects of Erdr1 in promoting Lgr5[+] ISC regeneration and crypt cell proliferation after radiation-induced injury, as well as in response to mucosal damage from DSS, indicate that this factor may be exploited for therapeutic benefit during cancer therapy and in the context of mucosal ulcerations, such as seen in gastric ulcers and in human inflammatory bowel disease.

## Methods

**Mice**. C57BL/6, Lgr5-EGFP-IRES-creERT2[32,51], and Myd88[−/−] mice were obtained from The Jackson Laboratory and housed in SPF conditions. TLR4[−/−] mice were provided by A.T.G. GF C57BL/6 mice were maintained under GF conditions in Park Bioservice isolators. Unless otherwise stated, mice were used at 6–8 weeks of age. Experiments were carried out using age- and sex-matched groups and complied with all relevant ethical regulations for animal testing and research. Animal studies were approved by the Institutional Animal Care and Use Committee of Georgia State University.

**exGF model**. GF C57BL/6 mice were born and raised in GF conditions until weaning time point at post birth day 21. At this time, GF mice were conventionalized by co-housing with SPF mice for 3 weeks. These mice are referred to as exGF mice.

**Organoid culture**. Isolation of crypt cells and organoid cultures were performed as previously described[32]. Organoids from the small and large intestine were established from freshly isolated tissue. Isolated small and large intestine were cut longitudinally and washed with cold PBS. Crypts were incubated for 1 h at room temperature in Gentle Cell Dissociation Reagent (Stemcell Technologies) and released from tissue by pipetting. Crypts were then passed through a 70 µm cell strainer and the crypt fraction was enriched by centrifugation. Crypts were subsequently embedded with matrigel and plated. After polymerization of matrigel, culture media (Stemcell Technologies) was added and refreshed every 2 days. For recombinant Erdr1 treatment, organoids were cultured in culture medium with or without recombinant Erdr1 (1 µg/mL). For flow cytometry analysis, organoids were vigorously pipetted for mechanical disruption, then dissociated using TrypLE (37 °C) and finally passed through a 70 µm cell strainer. Where applicable, cells were directly stained, fixed and permeabilized with Foxp3 Transcription Factor Fixation/Permeabilization kit (Invitrogen) and stained with Ki-67 antibody (12-5698-82, Invitrogen, 1:200 dilution). For Annexin V/PI staining, organoids were stained using the Dead Cell Apoptosis Kit with Annexin V FITC and PI (V13242, Invitrogen) according to the manufacturer's protocols and analyzed by flow cytometry. Human colonic enteroids (colonoids) were compliant with all relevant ethical regulations for work with human participants, including obtaining informed consent. All human colon sample collection was performed in accordance with the University of Michigan Institutional Review Board regulations. Isolated colonoids were resuspended in matrigel and cultured in growth media (50% L-WRN conditioned media:50% Advanced DMEM/F-12, 10% FBS, 2 mM GlutaMax, 10 mM HEPES, N-2 media supplement, B-27 Supplement, 1 mM N-Acetyl-L-cysteine, 50 ng/mL huEGF,

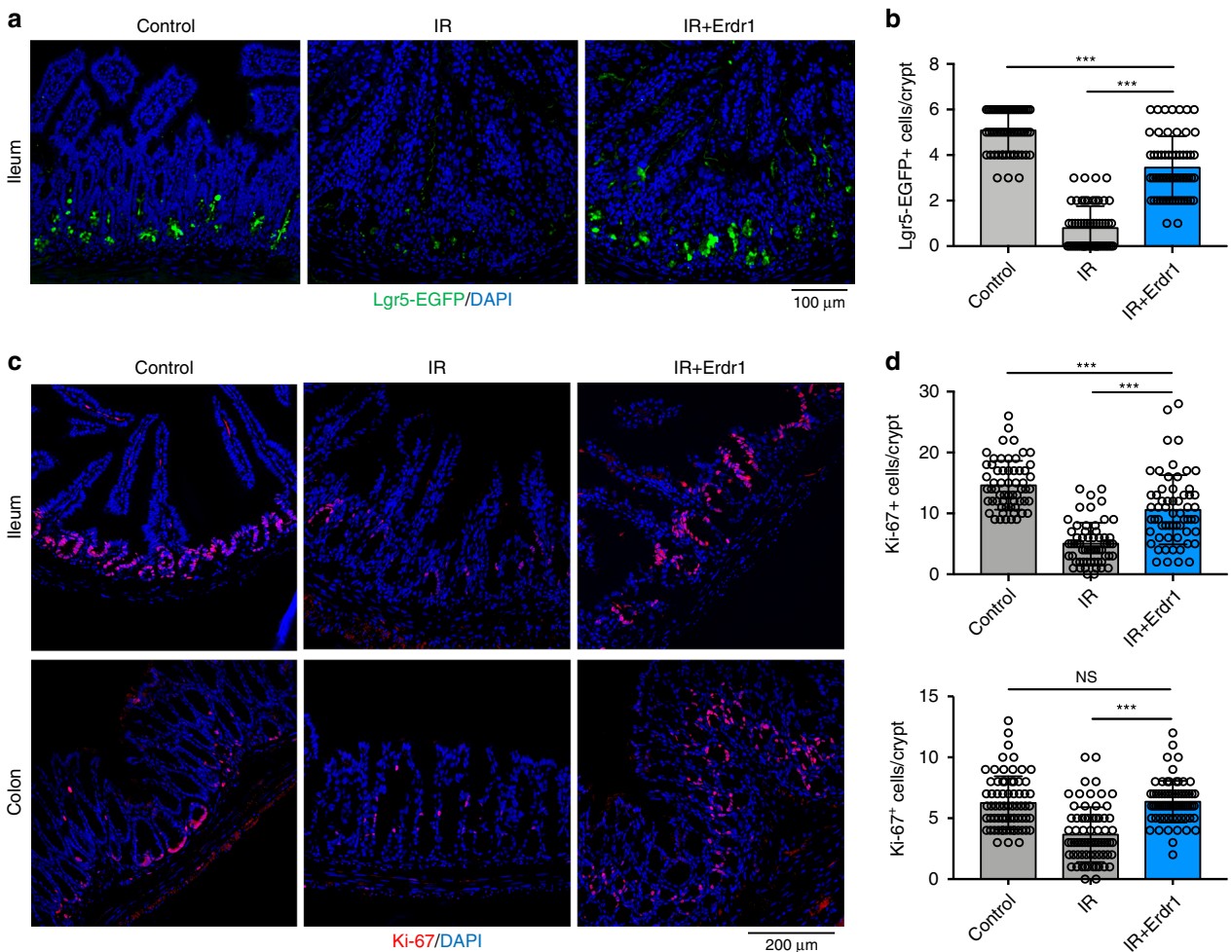

**Fig. 7 Erdr1 promotes the regeneration of Lgr5+ ISCs following radiation-induced injury. a** Representative images of EGFP staining in ileum sections from Lgr5-EGFP reporter mice housed in SPF conditions. Mice were injected ±Erdr1 daily following 10 Gy irradiation. **b** Quantification of Lgr5-EGFP+ cells per crypt. $n = 60$; representative of three experiments. **c** Representative images of Ki-67 staining in ileum and colon sections (pink, Ki-67; blue, DAPI). **d** Quantification of Ki-67 positive cells per crypt from the ileum (top) and colon (bottom). $n = 60$; representative of three experiments. All data are presented as mean ± SD; *$P < 0.05$, **$P < 0.01$, ***$P < 0.001$ by one-way ANOVA with Tukey's multiple comparison test.

100 units/mL Penicillin, 0.1 mg/mL streptomycin, 500 nM A83-01, 10 μM SB202190, 10 mM Nicotinamide, 10 nM Gastrin). Media was replaced every other day, and colonoid cultures were maintained via passaging one time per week. To generate 2D monolayers, colonoids grown as described above were spun out of matrigel and dissociated into a single cell suspension using Trypsin/EDTA in collagen coated 48 well tissue culture plates according to published protocols[50]. Following 1 day in complete growth media cells were switched to differentiation media (growth media minus Wnt3a, R-Spondin, Nicotinamide and SB202190 and with a 50% reduction in Noggin) for 4–5 days of differentiation into monolayers of colonoid epithelium.

**Cell culture**. Mode-K cells were cultured with DMEM in presence of 10% FBS, 100 IU penicillin, 100 μg/ml streptomycin. HT-29 cells were cultured with McCoy's Medium in presence of 10% FBS, 100 IU penicillin, and 100 μg/ml streptomycin.

**Scratch-wound assay**. For scratch-wound assays, cells were cultured in a 24-well plate until they achieved confluence. Six hours prior to making the scratch fresh medium ± Erdr1 (1 μg/mL) was added and subsequently a scratch across the monolayer was made using a P200 pipette tip. For Erdr1 knockdown experiments, sub-confluent Mode-K cells were transfected with anti-Erdr1 siRNA (L-053706-01, Dharmacon) or control siRNA (D-001810-1, Dharmacon) using lipofectamine 2000 (Invitrogen) 24 h prior to scratch-wounding. Wound closure was measured by ImageJ software. For time-lapse imaging of wound closure, movies were captured using an EVOS FL Auto imaging system with humidified, on-stage incubator at 37 °C, 5% $CO_2$.

**Cell cycle analysis**. Cells stimulated Erdr1 were fixed with 70% ethanol for 3 h on ice. After washing with cold PBS, cells were stained with FxCycle PI/RNase Staining Solution (F10797, Invitrogen) and analyzed by flow cytometry.

**Recombinant Erdr1**. Generation of recombinant Erdr1 was performed by Gene-Script (NJ, US). Briefly, full length Erdr1 was cloned into the pET30a vector to generate N-terminal His-tagged Erdr1. E.coli was transformed with this plasmid and cultured in medium containing ampicillin. IPTG was introduced for protein induction. Erdr1-His was purified on a nickel column and purity was >90% as confirmed by SDS-PAGE and MALDI-TOF. Potential endotoxin was removed using High Capacity Endotoxin Removing Spin Columns (88274, ThermoFisher).

**anti-Erdr1 polyclonal antibody**. Generation of anti-Erdr1 pAb was performed GeneScript (NJ, US). Briefly, rabbits were immunized with full length recombinant Erdr1 and subsequently purified using an Erdr1 affinity column. Potential endotoxin was removed using High Capacity Endotoxin Removing Spin Columns (88274, ThermoFisher).

**In situ hybridization and immunofluorescence staining**. For in situ hybridization and immunofluorescence staining, samples from small and large intestine were fixed in 10% formalin, embedded in paraffin and processed into 5 μm sections. In situ hybridization was performed using the RNAscope 2.5HD kit (Advanced Cell Diagnostics) following the manufacturer's protocol. We used an Erdr1 probe (465108, Advanced Cell Diagnosis) and a negative control probe (310043, Advanced Cell Diagnosis). TSA plus fluorescein, TSA plus Cyanine 3, and TSA plus Cyanine 5 (PerkinElmer) were used for detection. For BrdU staining, BrdU (423401, Biolegend) was dissolved in PBS to 5 mg/mL and 200 μL was injected intraperitoneally 6 h before euthanization to label proliferating cells. For immunofluorescence staining, tissue sections were permeabilized with 0.3% TritonX-100 and blocked in 3% BSA for 1 h at room temperature prior to incubation overnight with anti-Ki-67 antibody (ab15580, Abcam, 1:500 dilution), anti-BrdU antibody (ab6326, abcam, 1:250 dilution) or anti-GFP antibody (GFP-1020, Aves, 1:1000

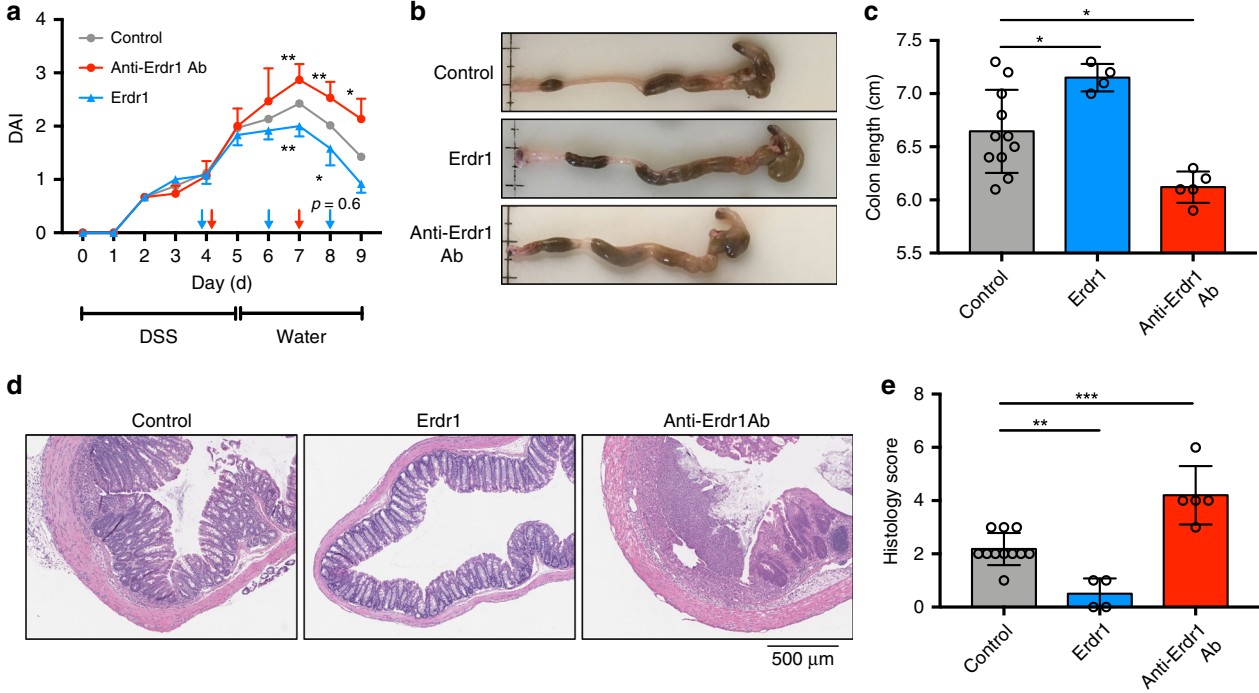

**Fig. 8 Erdr1 promotes recovery from dextran sulfate sodium (DSS)-induced colitis. a** DAI score of mice treated for 5 days with DSS, followed by normal drinking water for 4 days. Erdr1 was administered at day 4, 6, and 8; anti-Erdr1 antibody was administrated at day 4 and day 7. **b** Colon images from mice treated in **a**. **c** Colon length from mice treated in **a**. **d** Representative H&E stained sections and **e** histology scores from mice as treated in **a**. $n = 11$ (control), $n = 4$ (Erdr1), $n = 5$ (anti-Erdr1 Ab); representative of two experiments. All data are presented as mean ± SD; *$P < 0.05$, **$P < 0.01$, by one-way ANOVA with Tukey's multiple comparison test.

dilution) at 4 °C. Alexa Fluor 488-conjugated anti-rabbit IgG secondary antibody (A21206, Invitrogen, 1:1000 dilution), Alexa Fluor 647-conjugated anti-Rat IgG secondary antibody (A21247, Invitrogen, 1:1000 dilution) or Alexa Fluor 647-conjugated anti-chicken IgY secondary antibody (A21449, Invitrogen, 1:1000 dilution) were added for 1 h. Images of sections were obtained using a Zeiss LSM800 confocal microscope.

**Chromatin immunoprecipitation (ChIP).** ChIP assays were performed using the SimpleChIP Plus Enzymatic Chromatin IP kit (9004 S, Cell Signaling). Briefly, chromatin DNA was obtained from colon tissue or cultured organoids after fixation with formaldehyde and fragmented using micrococcal nuclease and sonication to generate lengths of ~150–900 bp. Digested chromatin was then incubated with control IgG (2729, Cell Signaling) or anti-acetyl-histone H3 Ab (06-599, EMD Millipore, 1:150 dilution) and precipitated using protein G agarose beads (9007, Cell Signaling). ChIP DNA was purified and subjected to qPCR using iTaq Universal SYBR Green Supermix (1725124, Bio-Rad) on StepOnePlus real-time PCR system (Applied Biosystems). Specific primers were designed targeting 1000 bp upstream of transcriptional starting site of Erdr1: F, 5'-GATCGCCATGTGCTCG C-3'; R, 5'-GGAGGCCGAGTCCGAT-3'.

**Radiation-induced injury.** For radiation-induced injury, mice received whole body X-ray radiation (10 Gy) and were injected with Erdr1 (6 μg/mice) daily on days 0, 1, and 2. At day 3 post radiation, intestinal tissue was collected and stained with Ki-67 and EGFP (as a surrogate of Lgr5 expression). For TUNEL assays, paraffin embedded tissue sections were stained using the Click-iT TUNEL Assay for In Situ Apoptosis Detection Alexa Fluor 647 dye (C10619, Invitrogen).

**DSS model of colonic damage.** Mice were treated with 3% (wt/vol) DSS (MP Biomedicals; molecular weight: 36,000–50,000) in the drinking water for 5 days and then switched to normal drinking water. Mice were monitored daily for signs of disease with score of 0–4 assigned for weight loss, stool consistency, and presence of blood in stool. The individual scores were added and the average score was DAI. For anti-Erdr1 Ab treatment, mice were injected (i.p.) on day 4 and 7 with 160 μg of anti-Erdr1 Ab and analyzed at day 9.

**Histology.** Colons were fixed with 10% formalin and embedded by paraffin. Tissue was cut into 5 μm sections and stained with hematoxylin/eosin. The degree of inflammation and epithelial damage was scored with average of two parameters include immune cell infiltration (0–3) and intestinal architecture (0–3).

**RNA isolation and qPCR.** Total RNA was isolated from cells and tissues using a Qiagen RNeasy Mini Kit and QIAcube with on column DNase digestion. cDNA was generated using the Hight-Capacity cDNA Reverse Transcriptional Kit (Applied Biosystems). qPCR was performed with SYBR Green on a StepOnePlus real-time PCR system (Applied Biosystems) and gene expression was normalized to *Gapdh*. Primers used were:

*Erdr1* (F, 5'-TGATGTCACCCACGAAAGCA-3'; R, 5'-TTCCTCCGTGAGAA TCGCTC-3')

*Lgr5* (F, 5'-GTGGACTGCTCGGACCTG-3'; R, 5'-GCTGACTGATGTTGTTC ATACTGAG-3')

*Ascl2* (F, 5'-GTTAGGGGGCTACTGAGCAT-3'; R, 5'-GTCAGCACTTGGCA TTTGGT-3')

*Smoc2* (F, 5'-CCCAAGCTCCCCTCAGAAG-3'; R, 5'-GCCACACACCTGGA CACAT-3')

*Olfm4* (F, 5'-CTGCTCCTGGAAGCTGTAGT-3'; R, 5'-ACCTCCTTGGCCAT AGCGAA-3')

*Tnfrsf19* (F, 5'-GAGGCTGGGAAGACAGGGAA-3'; R, 5'-AAGCTAGTGGC TGAAAGGATGG-3')

*Axin2* (F, 5'-TACGAGGAAGACCCGCAGA-3'; R, 5'-GAGCAGGGAGTGG TACTGC-3')

*Myc* (F, 5'-CGCGATCAGCTCTCCTGAAA-3'; R, 5'GCTGTACGGAGTCGT AGTCG-3')

*Sox9* (F, 5'-TGAAGAACGGACAAGCGGAG-3'; R, 5'-CAGCTTGCACGTCG GTTTTG-3')

*Ephb2* (F, 5'-TACAACGCCACGGCCATAAA-3'; R, 5'-CCAACGATGAGGG GTAGCTT-3')

*Hes1* (F, 5'-CCAGCCAGTGTCAACACGA-3'; R, 5'-AATGCCGGGAGCTAT CTTTCT-3')

*Yap1* (F, 5'-CGGCAGTCCTCCTTTGAGAT-3'; R, 5'-TTCAGTTGCGAAAG CATGGC-3')

*Cyr61* (F, 5'-CAGTGCTGTGAAGAGTGGGT-3'; R, 5'-GCGTGCAGAGGGT TGAAAAG-3')

*Ctgf* (F, 5'-AGGGCCTCTTCTGCGATTTC-3'; R, 5'-CTTTGGAAGGACTCA CCGCT-3')

*Gapdh* (F, 5'-TGCACCACCAACTGCTTAG-3'; R, 5'-GGATGCAGGGATGA TGTTC-3')

**RNAseq**. Total RNA was collected using Qiagen RNeasy Mini Kit (QIAGEN) from two mice in each group. RNA quality was assessed by nanodrop, agarose gel electrophoresis and using an Agilent 2100 Bioanalyzer. The mRNA-seq experiments were performed by Novogene (Beijing, China). We mapped the sequencing reads to the reference mouse genome GRCm38/mm10 using SYAR v2.5. Gene expression analysis was performed using HTSeq v0.6.1. Volcano plots were generated using R.

**TOP/FOP assay**. SKCO-15 colonic epithelial cells were seeded in 48-well tissue culture plates (60,000 cells per well) and transiently transfected with a β-catenin reporter containing 3 TCF/LEF-binding sites upstream of the luciferase reporter (TOP-Flash plasmid) or a negative control, FOP-Flash, which contains three TCF/LEF-mutated binding sites upstream of the luciferase reporter (Upstate Biotechnology). Transfections were carried out using Lipofectamine 2000 (Invitrogen Life Technologies). Seventy-two hours following transfection and Erdr1 treatment, Luciferase activity was measured in cell lysates using the Dual Luciferase Reporter Assay System (Promega) in the GloMax 96 Luminometer.

**Reporting summary**. Further information on research design is available in the Nature Research Reporting Summary linked to this article.

## Data availability

RNAseq raw data have been deposited in the European Nucleotide Archive: PRJEB35428. The source data underlying Figs. 1c–d, 3a–b, d–f, 4a, c–d, 5a–b, 6b, d, f–g, i, 7b, d, 8a, c, e, and Supplementary Figs. 1, 4b–d, 5b–c, 6–9, 11a–b, 12, 13b–d, f, 14b, 15b, 16b are provided as a Source Data.

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

## Author contributions

H.A. and T.L.D. conceived the idea for this project and designed the experiments. H.A. performed most of the experiments and analyzed the data. B.C analyzed RNAseq data. M.Q. and J.C.B. performed TOP/FOP assay and human organoid culture. A.H., V.L.N., and E.V. provided technical support. B.C., A.N., D.M., and A.T.G. provided reagents, mice and critical discussion. The manuscript was written by H.A and T.L.D.

## Competing interests

The authors declare no competing interests.
