## [Peer Review File · Nature Communications]

Reviewers' comments:

Reviewer #1 (Remarks to the Author):

The manuscript by Abo and colleagues entitled "Erythroid differentiation regulator-1 is induced by microbiota in early life and drives intestinal stem cell proliferation and regeneration" provides new data showing that *Erdr1* is expressed in the intestinal stem/TA compartment in mice in early life in SPF conditions, but not in germ-free conditions. Interestingly, *Erdr1* is not induced to normal levels even in ex-germ-free mice. Data is presented showing that *Erdr1* induced Wnt signaling, increased Lgr5+ ISCs and organoid growth. Notably, *Erdr1* accelerated wound closure in scratch-wound assays in vitro and increased Lgr5+ ISC regeneration following radiation and promoted recovery of mice from DSS-induced colonic injury. Alternatively, treatment of mice with anti-*Erdr1* antibody was shown to delay recovery from DSS-induced colonic injury.

Overall, this is a timely and important area of investigation. The manuscript is nicely written and notably thorough in experimentation with compelling in vitro and in vivo data. Addressing the following points will further strengthen this manuscript:

1. The authors should confirm that recombinant *Erdr1* has the expected properties using SDS-PAGE and/or mass spec.
2. The authors should show data using organoids generated from TLR4-deficient mice to verify that effects are not influenced by LPS.
3. The authors should perform qPCR of Notch target genes, such as *Hes1*, to test whether *Erdr1* activates Notch signaling in addition to Wnt signaling pathways.
4. The authors should examine if *Erdr1* regulates YAP-1 expression, given that YAP-1 is important for intestinal regeneration.

Reviewer #2 (Remarks to the Author):

Denning and colleagues suggested that erythroid differentiation regulator-1 (*Erdr1*) is transactivated by microbiota at the early stage of development and positively modulates intestinal stem cell proliferation and regeneration. Overall, the authors' finding is very interesting in regards to its novelty, the crucial roles of microbiota in physiology and pathology of the intestine. However, such interest is dampened by the lack of mechanisms including how *Erdr1* expression is induced by microbiota or microbial metabolites and how *Erdr1* activates Lgr5+ intestinal stem cells (ISCs) possibly via Wnt/beta-catenin signaling.

Major comments

1. The recommendation is to include RNA-seq results as a supplementary table.
2. It may need justification for why *Erdr1* was chosen among others.
3. Fig. 1. Epigenetic regulation of *Erdr1*. This needs more discussion. Also, additional experiments using organoids treated with 5'-Aza-deoxycytidine or HDAC inhibitor may address some part of the potential mechanism of epigenetic regulation of *Erdr1*.
4. Wound healing assays (Fig. 6) seems somewhat irrelevant to the main topic, i.e., ISCs and intestinal regeneration given that *Erdr1* expression is specifically enriched in ISCs and TA cells. Any justification?
5. Fig. 7. To claim the proliferative/regenerative effects of *Erdr1* on Lgr5+ ISCs and TA cells upon ionizing radiation (IR), authors should show the Ki67 images in a time-dependent manner (after IR). Also, BrdU incorporation or migration assays will better address the effects of *Erdr1* on ISCs and TA

cells during regeneration. Additionally, apoptosis should be analyzed (e.g., cleaved caspase-3). These additional experiments are required because it is unclear whether Erdr1 affects the proliferation or protects from IR injury of ISCs and TA cells. This is also due to the co-treatment of Erdr1 with IR, which makes it difficult to assess the effects of Erdr1 on intestinal regeneration (radioprotection, prevention, or increased regeneration?), which might also be addressed by pre-treatment of Erdr1 followed by IR.

6. Fig. 7. Did GF or ExoGF mice treated with IR display the severe impairment in intestinal regeneration with lethal phenotype? If they did, a gain-of-function approach for rescue (Erdr1 administration) might further address the crucial roles of Erdr1 in intestinal regeneration. Conversely, as shown in Fig. 8, anti-Erdr1 ab can be used for a loss-of-function approach to corroborate this claim.

7. How does Erdr1 activate Wnt/beta-catenin signaling?

Minor comments

1. 'Intestinal epithelial cells are highly sensitive to DNA damage caused by radiation': This is inaccurate. Only proliferating cells (Lgr5+ ISCs and TA) are sensitive to radiation (Suh et al., Cell Reports 2017 PMID: PMC5138641).
2. The recommendation is to revise the references (#37, 38). For example, Kuo lab at Stanford or Capecchi's Bmi1+ ISCs.
3. Fig. 2. DAPI labeling is missing in images.
4. Fig. 4. Which mice were used to isolate IECs/organoids? Assumed exGF or GF?
5. Fig. 5. What is the expression level of Erdr1 in IEC cell lines?
6. Fig. 5. Erdr1-activated beta-catenin target genes. Please include beta-catenin IF images upon Erdr1 treatment.
7. Two IEC cell lines were used but for specific assays for each. Any justification?
8. Fig. 6. Why does Mode-K cell express the endogenous Erdr1 although this cell line is not Lgr5+ ISCs or TA cells?
9. Fig. 7. Are the results from GF/ExoGF mice or SPF mice?

Reviewer #3 (Remarks to the Author):

Abo et al showed Erdr1 expressed Intestinal stem cells and TA cells in SPF mice, but not GF, eGF mice. Erdr1 expression increased total number of stem cells and Erdr1 is beneficial for the recovery tissue damage. Mouse model was elegant but there are issues to be solved.

First, Authors showed Erdr1 was expressed SPF mice, however, previous report SPF CD4 T cell reduced the expression level of Erdr1 compared to GF condition (Weis AM et al. Gut Microbes 2018.). Authors need to compare isolated EC cells between GF, eGF and SPF condition. If possible, to analyze Erdr1 expression of CD4 or NK cells in each condition is helpful to clarify their findings.

Second, authors demonstrated microbe is essential to increase Erdr1 expression in SPF mice, but exGF mice did not increase Erdr1 expression. Other papers demonstrated Erdr1 expression is decreased by TLR2-Myd88 pathway (Soto Ret al PNAS 2017). Authors should show expression of Erdr1 in intestinal epithelial cells is regulated by TLR-Myd88 or not. Also, recently Nabhani ZA showed microbe-IFN γ signaling at weaning age is essential for the low susceptibility (Nabhani ZA et al. Immunity 2019). Authors need to address the reason exGF mice express less Erdr1 under the microbe rich condition. Is specific microbe or cytokine essential for the upregulation of Erdr1 in stem cells?

Third, authors did not show the precise mechanism of Erdr1 control the number of stem cells. Previous report showed Erdr1 protein induce apoptosis by increased Fas/ Caspase8 and 3 on T cells (Weis AM et al.). Erdr1 signal is Fas dependent manner. Authors data is completely opposite result from previous report. These results might be Erdr1 unknown receptor did not express the intestinal stem cells

compared to the T/ NK cells. The receptor signal is further study but at least, authors need to address apoptosis factor in vitro and vivo. Also Edr1KO organoid assay should be important to understand for Erd1-Wnt signal.

Overall, the finding that microbe regulate Erd1 expression of intestinal stem/TA cells is potentially interesting, however current results cannot satisfy the readers. Authors need to address new insight/ pathway that is not preciously reported.

Minor:

Figure 2 : Authors need to show the Erd1 staining was fine by staining CD4 T cells in GF mice. See the major comments.

Figure 3: Authors should show pictures of entire culture well on Figure3c. Organoid experiment showed short term culture result, how about long term culture?

Figure 5: Did you check the expression level of Edr1 in SKCO15 cells?

Figure 6: Add 2D functional assay by using Edr1KO organoid.

Figure 7,8 : Authors did not show the adding Erd1 directly act on EC cells but not other immune cells. Previous report showed downreguation of Erd1 increased the number of Th17 pathogenic cells in EAE model. Authors need to eliminate the possibility of affecting other immune cells, by using RAGKO mice or with anti CD4/ anti NK1.1 antibody.

Reviewer #1 (Remarks to the Author)

The manuscript by Abo and colleagues entitled “Erythroid differentiation regulator-1 is induced by microbiota in early life and drives intestinal stem cell proliferation and regeneration” provides new data showing that Erdr1 is expressed in the intestinal stem/TA compartment in mice in early life in SPF conditions, but not in germ-free conditions. Interestingly, Erdr1 is not induced to normal levels even in ex-germ-free mice. Data is presented showing that Erdr1 induced Wnt signaling, increased Lgr5+ ISCs and organoid growth. Notably, Erdr1 accelerated wound closure in scratch-wound assays in vitro and increased Lgr5+ ISC regeneration following radiation and promoted recovery of mice from DSS-induced colonic injury. Alternatively, treatment of mice with anti-Erdr1 antibody was shown to delay recovery from DSS-induced colonic injury.

Overall, this is a timely and important area of investigation. The manuscript is nicely written and notably thorough in experimentation with compelling in vitro and in vivo data. Addressing the following points will further strengthen this manuscript:

1. The authors should confirm that recombinant Erdr1 has the expected properties using SDS-PAGE and/or mass spec.
2. The authors should show data using organoids generated from TLR4-deficient mice to verify that effects are not influenced by LPS.
3. The authors should perform qPCR of Notch target genes, such as Hes1, to test whether Erdr1 activates Notch signaling in addition to Wnt signaling pathways.
4. The authors should examine if Erdr1 regulates YAP-1 expression, given that YAP-1 is important for intestinal regeneration.

Authors point-by-point response to Reviewer #1

Reviewer 1, Comment 1: *The authors should confirm that recombinant Erdr1 has the expected properties using SDS-PAGE and/or mass spec.*

Author's response to Reviewer 1, Comment 1: We thank Reviewer 1 for this comment. We confirmed the properties of recombinant Erdr1 using SDS-PAGE. These data are now shown in Supplementary Figure. 3.

Reviewer 1, Comment 2: *The authors should show data using organoids generated from TLR4-deficient mice to verify that effects are not influenced by LPS.*

Author's response to Reviewer 1, Comment 2: We appreciate this important recommendation. In Supplementary Figure 4 we confirmed that organoids generated from TLR4-deficient mice respond similarly to those generated from WT mice. These data strongly suggest that results of Erdr1 treatment are not due to LPS contamination. Further,

all of our Erdr1 preparations were passed over an endotoxin removal column as detailed in the methods section.

Reviewer 1, Comment 3: *The authors should perform qPCR of Notch target genes, such as Hes1, to test whether Erdr1 activates Notch signaling in addition to Wnt signaling pathways.*

Author's response to Reviewer 1, Comment 3: We thank Reviewer 1 for this insightful comment. We performed qPCR analysis of *Hes1* in LI organoid stimulated with Erdr1 and observed no change in *Hes1* mRNA expression by Erdr1 treatment compared to control (Supplementary Figure 10a). These data indicate that enhanced organoid growth driven by Erdr1 is associated with induction of Wnt- but not Notch-related signaling genes.

Reviewer 1, Comment 4: *The authors should examine if Erdr1 regulates YAP-1 expression, given that YAP-1 is important for intestinal regeneration.*

Author's response to Reviewer 1, Comment 4: We thank Reviewer 1 for this comment. We performed qPCR analysis of *Yap1* and the target genes *Cyr61* and *Ctgf* in LI organoid stimulated with Erdr1 (Supplementary Figure 10b). We observed no change in expression of these genes following Erdr1 treatment as compared to controls. These data suggest that increased organoid growth is not associated with induction of Yap/Taz-related signaling genes.

Reviewer #2 (Remarks to the Author)

Denning and colleagues suggested that erythroid differentiation regulator-1 (Erdr1) is transactivated by microbiota at the early stage of development and positively modulates intestinal stem cell proliferation and regeneration. Overall, the authors' finding is very interesting in regards to its novelty, the crucial roles of microbiota in physiology and pathology of the intestine. However, such interest is dampened by the lack of mechanisms including how Erdr1 expression is induced by microbiota or microbial metabolites and how Erdr1 activates Lgr5+ intestinal stem cells (ISCs) possibly via Wnt/beta-catenin signaling.

Major comments

1. The recommendation is to include RNA-seq results as a supplementary table.
2. It may need justification for why Erdr1 was chosen among others.
3. Fig. 1. Epigenetic regulation of Erdr1. This needs more discussion. Also, additional experiments using organoids treated with 5'-Aza-deoxycytidine or HDAC inhibitor may address some part of the potential mechanism of epigenetic regulation of Erdr1.
4. Wound healing assays (Fig. 6) seems somewhat irrelevant to the main topic, i.e., ISCs and intestinal regeneration given that Erdr1 expression is specifically enriched in ISCs and TA cells. Any justification?
5. Fig. 7. To claim the proliferative/regenerative effects of Erdr1 on Lgr5+ ISCs and TA cells upon ionizing radiation (IR), authors should show the Ki67 images in a time-dependent manner (after IR). Also, BrdU incorporation or migration assays will better address the effects of Erdr1 on ISCs and TA cells during regeneration. Additionally, apoptosis should be analyzed (e.g., cleaved caspase-3). These additional experiments are required because it is unclear whether Erdr1 affects the proliferation or protects from IR injury of ISCs and TA cells. This is also due to the co-treatment of Erdr1 with IR, which makes it difficult to assess the effects of Erdr1 on intestinal regeneration (radioprotection, prevention, or increased regeneration?), which might also be addressed by pre-treatment of Erdr1 followed by IR.
6. Fig. 7. Did GF or ExoGF mice treated with IR display the severe impairment in intestinal regeneration with lethal phenotype? If they did, a gain-of-function approach for rescue (Erdr1 administration) might further address the crucial roles of Erdr1 in intestinal regeneration. Conversely, as shown in Fig. 8, anti-Erdr1 ab can be used for a loss-of-function approach to corroborate this claim.
7. How does Erdr1 activate Wnt/beta-catenin signaling?

Minor comments

1. 'Intestinal epithelial cells are highly sensitive to DNA damage caused by radiation': This is inaccurate. Only proliferating cells (Lgr5+ ISCs and TA) are sensitive to radiation (Suh et al., Cell Reports 2017 PMID: PMC5138641).
2. The recommendation is to revise the references (#37, 38). For example, Kuo lab at Stanford or Capecchi's Bmi1+ ISCs.
3. Fig. 2. DAPI labeling is missing in images.
4. Fig. 4. Which mice were used to isolated IECs/organoids? Assumed exGF or GF?
5. Fig. 5. What is the expression level of Erdr1 in IEC cell lines?
6. Fig. 5. Erdr1-activated beta-catenin target genes. Please include beta-catenin IF images upon Erdr1 treatment.
7. Two IEC cell lines were used but for specific assays for each. Any justification?
8. Fig. 6. Why does Mode-K cell express the endogenous Erdr1 although this cell line is not Lgr5+ ISCs or TA cells?
9. Fig. 7. Are the results from GF/ExoGF mice or SPF mice?

Authors point-by-point response to Reviewer #2

Major comments

Reviewer 2, Comment 1: *The recommendation is to include RNA-seq results as a supplementary table.*

Author's response to Reviewer 2, Comment 1: We thank Reviewer 2 for this constructive comment. We have provided RNA-seq results as Supplementary Table 1.

Reviewer 2, Comment 2: *It may need justification for why Erdr1 was chosen among others.*

Author's response to Reviewer 2, Comment 2: We thank Reviewer 2 for this important comment. We selected Erdr1 based on three reasons: First, Erdr1 is one of the most differentially expressed genes between SPF mice and exGF or GF mice. Second, Erdr1 expression level in SPF mice was notably high while it was nearly undetectable in GF and exGF mice based on FPKF values. Third, the function of Erdr1 is still unclear in the gut is just beginning to emerge (Soto et al, PNAS 2017) and is clearly a candidate for further investigation.

Reviewer 2, Comment 3: *Fig. 1. Epigenetic regulation of Erdr1. This needs more discussion. Also, additional experiments using organoids treated with 5'-Azadeoxycytidine or HDAC inhibitor may address some part of the potential mechanism of epigenetic regulation of Erdr1.*

Author's response to Reviewer 2, Comment 3: We thank Reviewer 2 for this insightful comment. We have now further discussed our data related to the epigenetic regulation of Erdr1 on page 11. Erdr1 is initially expressed in utero in SPF mice and this expression is associated with the presence of microbiota in a temporally controlled window. As such, colonization of GF mice with SPF microbiota is insufficient to induce Erdr1 in adult exGF mice. These data underscore the likelihood that Erdr1 expression involves a microbiota-dependent induction/stimulus event in utero or in the early postnatal phase and if this does not take place then gene repression may take place via reduced histone H3 acetylation. Our ChIP analyses (Figure 1d; Figure 3b) demonstrate that reduced histone H3 acetylation in GF and exGF mice was associated with reduced Erdr1 expression. Given the complexity of epigenetic regulation, we expect to analyze acetylation, methylation, phosphorylation, etc in future studies.

Reviewer 2, Comment 4: *Wound healing assays (Fig. 6) seems somewhat irrelevant to the main topic, i.e., ISCs and intestinal regeneration given that Erdr1 expression is specifically enriched in ISCs and TA cells. Any justification?*

Author's response to Reviewer 2, Comment 4: We thank Reviewer 2 for allowing us to justify this line of experimentation. Erdr1 is expressed predominantly by ISCs and TA cells, however the functions of Erdr1 may be in an autocrine, paracrine, and/or endocrine fashion. In Figure 6 we examined the role of Erdr1 in a paracrine setting by adding exogenous Erdr1 or by inhibiting endogenous Erdr1 when using intestinal epithelial cell lines. These experiments demonstrate that while Erdr1 is expressed by ISCs and TA cells it may affect neighboring intestinal epithelial cells by enhancing proliferation and wound closure. Of note, these data do not exclude an autocrine role for Erdr1 and in fact such an autocrine role for Erdr1 on ISCs is likely as evidenced by enhanced Lgr5 ISCs upon Erdr1 treatment of organoids (Figure 4) and following ionizing radiation (Figure 7).

Reviewer 2, Comment 5: *Fig. 7. To claim the proliferative/regenerative effects of Erdr1 on Lgr5+ ISCs and TA cells upon ionizing radiation (IR), authors should show the Ki67 images in a time-dependent manner (after IR). Also, BrdU incorporation or migration assays will better address the effects of Erdr1 on ISCs and TA cells during regeneration. Additionally, apoptosis should be analyzed (e.g., cleaved caspase-3). These additional experiments are required because it is unclear whether Erdr1 affects the proliferation or protects from IR injury of ISCs and TA cells. This is also due to the co-treatment of Erdr1 with IR, which makes it difficult to assess the effects of Erdr1 on intestinal regeneration (radioprotection, prevention, or increased regeneration?), which might also be addressed by pre-treatment of Erdr1 followed by IR.*

Author's response to Reviewer 2, Comment 5: We thank Reviewer 2 for this comment. Now we have shown Ki-67, BrdU and TUNEL staining in a time-dependent manner after

IR (Supplementary Figure. 13-15). Ki-67 and BrdU images indicate that Erdr1 promotes proliferation on day 2 and 3 post radiation, while TUNEL staining revealed that apoptosis was not affected by administration of Erdr1. These data strongly suggest that Erdr1 has a role in promoting proliferation without affecting apoptosis following IR in the intestine.

Reviewer 2, Comment 6: *Fig. 7. Did GF or ExGF mice treated with IR display the severe impairment in intestinal regeneration with lethal phenotype? If they did, a gain-of-function approach for rescue (Erdr1 administration) might further address the crucial roles of Erdr1 in intestinal regeneration. Conversely, as shown in Fig. 8, anti-Erdr1 ab can be used for a loss-of-function approach to corroborate this claim.*

Author's response to Reviewer 2, Comment 6: We thank Reviewer 2 for this important comment. We did not perform end-stage lethality experiments due to strict IACUC regulations at our university. We agree that Erdr1 administration or blockade in a lethal model is likely to be informative, we hope that the reviewer can appreciate the ethical concerns by certain IACUCs regarding this type of experimentation. Additionally, performing IR on GF mice is technically impossible at our university since performing IR necessitates removal of mice from GF isolators upon which time they are immediately contaminated. Despite these limitations, we strongly believe that our data showing the potent effects of Erdr1 administration or blockade in SPF mice are even more relevant to defining the role of this factor in a physiological setting and in models of both IR and colitis (Figure 7 and 8).

Reviewer 2, Comment 7: *How does Erdr1 activate Wnt/beta-catenin signaling?*

Author's response to Reviewer 2, Comment 7: We thank Reviewer 2 for this important comment. To date, the receptor for Erdr1 has not been identified, therefore the precise mechanism via which Erdr1 activated Wnt/beta-catenin signaling remains obscure. Future studies identifying the receptor for Erdr1 should shed light into this important question.

Minor comments

Reviewer 2, Comment 1: *'Intestinal epithelial cells are highly sensitive to DNA damage caused by radiation': This is inaccurate. Only proliferating cells (Lgr5+ ISCs and TA) are sensitive to radiation (Suh et al., Cell Reports 2017 PMID: PMC5138641).*

Author's response to Reviewer 2, Comment 1: We thank Reviewer 2 for this comment. We have corrected this statement and added the appropriate reference.

Reviewer 2, Comment 2: *The recommendation is to revise the references (#37, 38). For example, Kuo lab at Stanford or Capecchi's Bmi1+ ISCs.*

Author's response to Reviewer 2, Comment 2: We thank Reviewer 2 for this comment. We have replaced reference #37 and 38 to Kuo lab paper.

Reviewer 2, Comment 3: *Fig. 2. DAPI labeling is missing in images.*

Author's response to Reviewer 2, Comment 3: We thank Reviewer 2 for this comment and apologize for the error. Figure 2 has been corrected.

Reviewer 2, Comment 4: *Fig. 4. Which mice were used to isolated IECs/organoids? Assumed exGF or GF?*

Author's response to Reviewer 2, Comment 4: We thank Reviewer 2 for this comment. We used SPF organoid from colon. We have now clarified this in the results section and figure legend.

Reviewer 2, Comment 5: *Fig. 5. What is the expression level of Erdr1 in IEC cell lines?*

Author's response to Reviewer 2, Comment 5: We thank Reviewer 2 for this comment. We performed qPCR for Erdr1 mRNA in Mode-K, HT-29 and SKCO15 cell lines. Mode-K cells express Erdr1 mRNA (Supplementary Figure 11), however, the expression in HT-29 and SKCO15 was undetectable despite using the same primers validated to work in human cells (Soto et al, PNAS 2017).

Reviewer 2, Comment 6: *Fig. 5. Erdr1-activated beta-catenin target genes. Please include beta-catenin IF images upon Erdr1 treatment.*

Author's response to Reviewer 2, Comment 6: We thank Reviewer 2 for this comment. We performed immunofluorescence staining on organoid cultures treated with Erdr1. As shown in Supplementary Figure 9, we detected translocation of activated beta-catenin in the presence Erdr1 treatment. These data are fully consistent with activation of WNT signaling pathway in Figure 5.

Reviewer 2, Comment 7: *Two IEC cell lines were used but for specific assays for each. Any justification?*

Author's response to Reviewer 2, Comment 7: We thank Reviewer 2 for allowing us to clarify our use of cell lines. In Figure 6 and Supplementary Figure 11 we used the Mode-K cell line for both Erdr1 administration as well as siRNA experiments. Mode-K is a widely used intestinal epithelial cell line in mouse studies. In Figure 5, we used SKCO15 cells as they have been optimized for use with the TOP/FOP luciferase reporter assay. In Supplementary Figure 12 we used HT29 for scratch wound assay as they have been widely used for this purpose in the literature.

Reviewer 2, Comment 8: *Fig. 6. Why does Mode-K cell express the endogenous Erdr1 although this cell line is not Lgr5+ ISCs or TA cells?*

Author's response to Reviewer 2, Comment 8: We thank Reviewer 2 for this comment. Mouse intestinal epithelial cell line Mode-K was established from the duodenum (Vidal et al., J Immunol Methods 1993). When Mode-K cell were established, primary intestinal

epithelial cells were cultured for 8 days before infecting SV40 large T antigen. During this 8-day culture, proliferating cells were selected. While clearly not prototypic ISCs or TA cells, whether these intestinal epithelial cells maintained any lineage relationship with ISCs or TA cells remains unclear.

Reviewer 2, Comment 9: *Fig. 7. are the results from GF/ExoGF mice or SPF mice?*

Author's response to Reviewer 2, Comment 9: We thank Reviewer 2 for this comment. We got results of Figure 7 from SPF mice. We have now clarified this in the results section and figure legend.

Reviewer #3 (Remarks to the Author)

Abo et al showed Edr1 expressed Intestinal stem cells and TA cells in SPF mice, but not GF, eGF mice. Edr1 expression increased total number of stem cells and Edr1 is beneficial for the recovery tissue damage. Mouse model was elegant but there are issues to be solved.

First, Authors showed Edr1 was expressed SPF mice, however, previous report SPF CD4 T cell reduced the expression level of Erd1 compared to GF condition (Weis AM et al. Gut microbe 2018.). Authors need to compare isolated EC cells between GF, eGF and SPF condition. If possible, to analyse Edr1 expression of CD4 or NK cells in each condition is helpful to clarify their findings.

Second, authors demonstrated microbe is essential to increase Erd1 expression in SPF mice, but exGF mice did not increase Erd1 expression. Other papers demonstrated Erd1 expression is decreased by TLR2-Myd88 pathway (Soto Ret al PNAS 2017). Authors should show expression of Erd1 in intestinal epithelial cells is regulated by TLR-Myd88 or not. Also, recently Nabhani ZA showed microbe-IFNg signaling at weaning age is essential for the low susceptibility (Nabhani ZA et al. Immunity 2019). Authors need to address the reason exGF mice express less Edr1 under the microbe rich condition. Is specific microbe or cytokine essential for the upregulation of Edr1 in stem cells? Third, authors did not show the precise mechanism of Edr1 control the number of stem cells. Previous report showed Erd1 protein induce apoptosis by increased Fas/ Caspase8 and 3 on T cells (Weis AM et al.). Erd1 signal is Fas dependent manner. Authors data is completely opposite result from precious report. These results might be Erd1 unknown receptor did not express the intestinal stem cells compared to the T/ NK cells. The receptor signal is further study but at least, authors need to address apoptosis factor in vitro and vivo. Also Edr1KO organoid assay should be important to understand for Erd1-Wnt signal.

Overall, the finding that microbe regulate Erd1 expression of intestinal stem/TA cells is potentially interesting, however current results cannot satisfy the readers. Authors need to address new insight/ pathway that is not preciously reported.

Minor:

Figure 2 : Authors need to show the Erd1 staining was fine by staining CD4 T cells in GF mice. See the major comments.

Figure 3: Authors should show pictures of entire culture well on Figure3c. Organoid experiment showed short term culture result, how about long term culture?

Figure 5: Did you check the expression level of Edr1 in SKCO15 cells?

Figure 6: Add 2D functional assay by using Edr1KO organoid.

Figure 7,8 : Authors did not show the adding Erd1 directly act on EC cells but not other

immune cells. Previous report showed downregulation of Erd1 increased the number of Th17 pathogenic cells in EAE model. Authors need to eliminate the possibility of affecting other immune cells, by using RAGKO mice or with anti CD4/ anti NK1.1 antibody.

Authors point-by-point response to Reviewer #3

Major comments

Reviewer 3, Comment 1: *Authors showed Edr1 was expressed SPF mice, however, previous report SPF CD4 T cell reduced the expression level of Erd1 compared to GF condition (Weis AM et al. Gut microbe 2018.). Authors need to compare isolated EC cells between GF, eGF and SPF condition. If possible, to analyse Edr1 expression of CD4 or NK cells in each condition is helpful to clarify their findings.*

Author's response to Reviewer 3, Comment 1: We thank Reviewer 3 for this comment. In Figure 2a and new Supplementary Figure 2 our data clearly demonstrate Erdr1 expression predominantly in the crypt region and TA zone of SPF, but not GF or exGF mouse intestine. It is important to clarify that these data are not inconsistent with those of Soto et al, PNAS, 2017, since that report did not compare the expression of Erdr1 in gut CD4 T cells from SPF and GF mice, but rather splenic CD4 T cells from SPF and GF mice. Further, we observed Erdr1 staining in the lamina propria region of the small intestine from SPF mice which is consistent with expression of Erdr1 by other cells types perhaps including CD4 T cells, as reported by Soto et al, PNAS, 2017 in the spleen. Interestingly, we did not observe detectable Erdr1 expression in the intestines of GF or exGF mice, so we did not pursue co-staining for specific cell types as that data would also be negative for Erdr1 expression. Collectively, these data suggest that microbiota-dependent regulation of Erdr1 expression may differ between the gut and periphery.

Reviewer 3, Comment 2: *Authors demonstrated microbe is essential to increase Erd1 expression in SPF mice, but exGF mice did not increase Erd1 expression. Other papers demonstrated Erd1 expression is decreased by TLR2-Myd88 pathway (Soto Ret al PNAS 2017). Authors should show expression of Erd1 in intestinal epithelial cells is regulated by TLR-Myd88 or not. Also, recently Nabhani ZA showed microbe-IFNg signaling at weaning age is essential for the low susceptibility (Nabhani ZA et al. Immunity 2019). Authors need to address the reason exGF mice express less Edr1 under the microbe rich condition. Is specific microbe or cytokine essential for the upregulation of Edr1 in stem cells?*

Author's response to Reviewer 3, Comment 2: We thank for Reviewer 3 for this important comment. To address this comment, we performed qPCR analysis using total SI and LI tissue from Myd88 KO mice. Knock out of Myd88 did not result in altered Erdr1 expression (Supplementary Figure 1). These data suggest that Erdr1 expression in the mouse intestine is independent of Myd88 signaling.

Further, the reason why exGF mice still express less Erdr1 even after being colonized under SPF conditions is because they have missed a critical window for microbiota

exposure in early life (prior to weaning) and explainable, at least in part, by epigenetic changes in the *Erdr1* promoter. As shown in Figure 1d and 3b, microbiota exposure prior to weaning promotes histone H3 acetylation on the *Erdr1* promoter region and this epigenetic change is associated with increased *Erdr1* expression. Since exGF mice were not exposed to microbiota (and their associated metabolites) before weaning, histone H3 acetylation on the *Erdr1* promoter and *Erdr1* expression was dramatically reduced. These findings indicate unknown mediators include metabolites produced by microbiota induce *Erdr1*. Study to find specific mediator which upregulates *Erdr1* is outside the scope of this study. We have now included these data in Supplementary Figure. 1.

While it is certainly of interest to define the specific microbe(s)/metabolite(s) and/or cytokine(s)/factor(s) that regulate *Erdr1* expression in ISCs, we hope the reviewer can appreciate that doing so is a massive undertaking that would last numerous years and is well beyond the scope of this initial study (currently 8 main figures and 16 supplementary figures) defining novel expression patterns of *Erdr1* in the gut, regulation of *Erdr1* by the microbiota, and functions of *Erdr1* in intestinal regeneration and colitis.

Reviewer 3, Comment 3: *Authors did not show the precise mechanism of *Erdr1* control the number of stem cells. Previous report showed *Erdr1* protein induce apoptosis by increased Fas/ Caspase8 and 3 on T cells (Weis AM et al.). *Erdr1* signal is Fas dependent manner. Authors data is completely opposite result from previous report. These results might be *Erdr1* unknown receptor did not express the intestinal stem cells compared to the T/ NK cells. The receptor signal is further study but at least, authors need to address apoptosis factor in vitro and vivo. Also *Erdr1*KO organoid assay should be important to understand for *Erdr1*-Wnt signal.*

Author's response to Reviewer 3, Comment 3: We thank for Reviewer 3 for this comment. We appreciate that previous papers have reported that *Erdr1* induces apoptosis. *Erdr1* has also been reported to be a survival factor during condition of cellular stress (Dormer et al, 2004). In addition, *Erdr1*-expressing stroma can promote cancer cell survival *in vitro* and cancer cell invasion *in vivo* (Mango et al, 2014). These papers and our finding strongly indicate that *Erdr1* play unique roles depending on the cell type, condition and tissue. We have now included a discussion to highlight the important and expanding biological functions of *Erdr1*.

In addition, we performed Annexin V/PI staining using organoids. *Erdr1* induced no change of apoptotic cells (Supplementary Figure 6). Also, at day 1 post radiation, *Erdr1* treated mice showed the same level of apoptosis as compared to control mice (Supplementary Figure 14). These *in vitro* and *in vivo* data indicate *Erdr1* does not induce apoptosis on IECs.

Lastly, we concur with Reviewer 3 that the study of *Erdr1* receptor is well beyond the scope of the current study.

Minor comments

Reviewer 3, Comment 1: *Figure 2: Authors need to show the Erd1 staining was fine by staining CD4 T cells in GF mice. See the major comments.*

Author's response to Reviewer 3, Comment 1: Please see response above to Reviewer 3, Comment 1. Our Erdr1 staining is very specific and robust in the intestine. ISCs and TA zone cells express robust levels of Erdr1 in the intestine of SPF mice and cells in the lamina propria (perhaps CD4 T cells) also express Erdr1 albeit at what appears to be lower levels.

Reviewer 3, Comment 2: *Figure 3: Authors should show pictures of entire culture well on Figure 3c. Organoid experiment showed short term culture result, how about long-term culture?*

Author's response to Reviewer 3, Comment 2: To avoid any concerns of non-representative regions of the well being used to generate organoid data, we selected 4 random regions and used those same regions for calculating data for all treatment groups.

Our data demonstrating that Erdr1 enhances organoid efficiency, budding, and surface area is consistent whether we add Erdr1 during the first week of organoid generation (short-term) or after passage during week 2 or week 3 (long-term).

Reviewer 3, Comment 3: *Figure 5: Did you check the expression level of Erdr1 in SKCO15 cells?*

Author's response to Reviewer 3, Comment 3: We thank for Reviewer 3 for this comment. We performed qPCR for Erdr1 mRNA in Mode-K, HT-29 and SKCO15 cell lines. Mode-K cells express Erdr1 mRNA (Supplementary Figure 11), however, the expression in HT-29 and SKCO15 was undetectable despite using the same primers validated to work in human cells (Soto et al, PNAS 2017).

Reviewer 3, Comment 4: *Figure 6: Add 2D functional assay by using Erdr1KO organoid.*

Author's response to Reviewer 3, Comment 4: We thank for Reviewer 3 for this comment. Erdr1 knockout mice are not available since the previous report failed to generate whole body KO and flox mice (Soto et al, PNAS, 2017).

Reviewer 3, Comment 5: *Figure 7,8: Authors did not show the adding Erd1 directly act on EC cells but not other immune cells. Previous report showed downregulation of Erd1 increased the number of Th17 pathogenic cells in EAE model. Authors need to eliminate the possibility of affecting other immune cells, by using RAGKO mice or with anti CD4/anti NK1.1 antibody.*

Author's response to Reviewer 3, Comment 5: We thank for Reviewer 3 for this comment and would like to emphasize that our manuscript is focused on defining novel expression patterns of Erdr1 in the gut, regulation of Erdr1 by the microbiota, and functions of Erdr1 in intestinal regeneration and colitis. We have indeed demonstrated

that Erdr1 has effects on intestinal epithelial cell lines (mouse and human; no immune cells present) and organoids (mouse and human; no immune cells present). These effects of Erdr1 are most definitely not due to effects on immune cells. We cannot and do not exclude a possible role for Erdr1 on other cell types beyond intestinal epithelial cells in vivo. We hope the reviewer can appreciate that requiring that we investigate the direct versus indirect effects of Erdr1 in vivo on numerous cells types is a massive undertaking that would last numerous years and is well beyond the scope of this initial study (currently 8 main figures and 16 supplementary figures).

REVIEWERS' COMMENTS:

Reviewer #1 (Remarks to the Author):

The authors have addressed all the concerns raised by the reviewers.

Reviewer #2 (Remarks to the Author):

Fig. S10. The nuclear beta-catenin does not look convincing, though there is a clear increase of beta-catenin protein. Please repeat IF staining with confocal microscope, which should give a better quality of image showing the nuclear beta-catenin.

Reviewer #3 (Remarks to the Author):

Abo et al showed the importance of Erdr1 in the stem cell and TA cells. The paper was nicely written and well-organized. I am delight to accept this manuscript.

Authors point-by-point response to editorial requests:

We have made all requested editorial changes throughout.

Authors point-by-point response to Reviewer #2

Reviewer 2, Comment 1: *Fig. S10. The nuclear beta-catenin does not look convincing, though there is a clear increase of beta-catenin protein. Please repeat IF staining with confocal microscope, which should give a better quality of image showing the nuclear beta-catenin.*

Author's response to Reviewer 2, Comment 1: We thank Reviewer 2 for this comment. These data are now shown in revised Supplementary Figure. 10. Green and red indicate β -catenin and nuclei, respectively, and clear colocalization (nuclear beta-catenin) is indicated by yellow.